# Unlocking Noise-Resistant Vision: Key Architectural Secrets for Robust Models Against Gaussian Noise

## Abstract

While the robustness of vision models is often measured, their dependence on specific architectural design choices is rarely dissected. We investigate why certain vision architectures are inherently more robust to additive Gaussian noise and convert these empirical insights into simple, actionable design rules. Specifically, we performed extensive evaluations on 1,174 pretrained vision models, empirically identifying four consistent design patterns for improved robustness against Gaussian noise: larger stem kernels, smaller input resolutions, average pooling, and supervised vision transformers (ViTs) rather than CLIP ViTs, which yield up to 506 rank improvements and 21.6%p accuracy gains. We then develop a theoretical analysis that explains these findings, converting observed correlations into causal mechanisms. First, we prove that low-pass stem kernels attenuate noise with a gain that decreases quadratically with kernel size and that anti-aliased downsampling reduces noise energy roughly in proportion to the square of the downsampling factor. Second, we demonstrate that average pooling is unbiased and suppresses noise in proportion to the pooling window area, whereas max pooling incurs a positive bias that grows slowly with window size and yields a relatively higher mean-squared error and greater worst-case sensitivity. Third, we reveal and explain the vulnerability of CLIP ViTs via a pixel-space Lipschitz bound: The smaller normalization standard deviations used in CLIP preprocessing amplify worst-case sensitivity by up to 1.91 times relative to the Inception-style preprocessing common in supervised ViTs. Our results collectively disentangle robustness into interpretable modules, provide a theory that explains the observed trends, and build practical, plug-and-play guidelines for designing vision models more robust against Gaussian noise.

## 1 Introduction

Vision models, implemented with deep neural networks, are now deployed across numerous fields, even in safety-critical applications ranging from medical imaging to autonomous driving. Their remarkable accuracy, however, conceals an uncomfortable fact: Performance can deteriorate when test images deviate—even slightly—from the training distribution (Hendrycks & Dietterich, 2019). Even light Gaussian noise can trigger misclassifications, and in autonomous vehicles, such brittleness can lead to life-threatening failures.

Recent studies have empirically discovered that the architectural design of deep neural networks strongly shapes their robustness to common image transformations. Specifically, Paul & Chen (2022); Bai et al. (2021); Naseer et al. (2021) observed that vision transformers (ViTs) often degrade less than previous convolutional networks, such as residual networks (ResNets), under various corruptions. Although promising results with ViTs have been reported, such studies typically treat each architecture as a whole, leaving unanswered which specific internal choices contribute to gains in robustness.

In this study, we dissect the robustness of vision models under Gaussian noise, showing that specific micro-architectural choices are key factors in determining robustness. We performed extensive experiments on available vision models from the `timm` library (Wightman, 2019), as well as controlled experiments; our empirical meta-analysis compares architectures pairwise within the vision models, which enables us to isolate the effect of each micro-architectural factor, thereby revealing four interesting design patterns in architectures that improve robustness against Gaussian noise:

- **Larger** stem kernels, such as larger patch sizes in ViTs, rather than **smaller** ones,
- **Smaller** input resolutions, such as $224^2$, rather than **larger** ones, such as $384^2$,
- **Average** pooling, rather than **max** pooling, and
- **Supervised** learning ViTs, rather than **CLIP** ViTs.

Extending these empirical observations, we also derive several theoretical results that account for the differences in these choices. Specifically, we prove that noise gain decays quadratically with the stem kernel size and that downsampling after anti-alias filtering yields analogous gains (Section 4). Then we analyze Gaussian-noise error formulas for both pooling operators, showing that average pooling is unbiased with decreased variance, whereas max pooling incurs a positive bias and a higher mean-squared error (Section 5). Finally, we demonstrate that the vulnerability of CLIP ViTs is primarily caused by the choice of mean-std normalization, whose effect is proven with Lipschitz bounds (Section 6).

## 2 RELATED WORK

**Robustness literature and positioning of this study.** Robustness to common corruptions is typically evaluated using ImageNet-C (Hendrycks & Dietterich, 2019). A consistent observation across studies is that ViTs often degrade less than CNNs do under such corruptions (Paul & Chen, 2022; Bai et al., 2021; Naseer et al., 2021). However, most prior comparisons treat architectures as monolithic families or vary training recipes, making it hard to isolate which micro-architectural choices drive robustness. Furthermore, multiple corruptions, such as brightness changes and blur, are mixed in. In contrast to these complex corruptions and architectures, we design a systematic evaluation protocol to isolate the effect of each micro-architectural factor. Furthermore, we select Gaussian noise due to its approximation of aggregate perturbations by the central limit theorem and its prevalence in real-world imaging, such as sensor readout and thermal noise. To this end, our experiments disentangle four architectural choices across pretrained models and controlled settings, enabling clean attribution. Our findings align with prior results on robustness studies (Paul & Chen, 2022; Boureau et al., 2010) and add causal, quantitative explanations. The parts below review related work that corresponds to the empirical design patterns we identified for enhancing Gaussian noise robustness.

**Anti-aliasing, kernels, and resolution.** Anti-aliased downsampling is known to reduce high-frequency sensitivity and improve stability (Zhang, 2019; Zou et al., 2023), and analogous ideas have been explored for ViTs (Qian et al., 2021). Complementing these studies, we provide explicit scaling laws: The output noise energy decays quadratically with the stem kernel size and the anti-aliased downsampling factor, explaining why larger stem kernels and smaller input resolutions improve robustness.

**Pooling under additive noise.** Classical analysis shows that average pooling is unbiased with variance reduction, whereas max pooling introduces a positive bias under Gaussian noise (Boureau et al., 2010); recent studies further clarify when max pooling aids invariance despite worse noise behavior (Matoba et al., 2023). We extend this line and empirically verify the predicted advantage of average pooling over max pooling across multiple datasets.

**Normalization, CLIP preprocessing, and Lipschitz sensitivity.** Vision models employ specific per-channel mean-std preprocessing, which, according to Lipschitz-based robustness theory (Virmaux & Scaman, 2018; Gouk et al., 2021; Tsuzuku et al., 2018), directly rescales pixel-space sensitivity. We make this

connection explicit: Smaller channel standard deviations enlarge the end-to-end Lipschitz bound, predicting greater worst-case and mean-squared sensitivity to additive noise.

## 3 WHY GAUSSIAN NOISE?

Our study focuses on robustness against additive Gaussian noise, which essentially captures comparable or even worst-case robustness against common image corruptions. Indeed, additive Gaussian noise is the least favorable among all perturbations whose covariance is spectrally bounded, which makes it a conservative measure of robustness.

**Setup.** Let $x \in [0,1]^d$ be an image, and let $f : \mathbb{R}^d \to \mathbb{R}^k$ be a vision model. A corruption produces $x' = \mathcal{C}(x, \xi)$ with $\Delta := x' - x$, and we write $\delta_f := f(x + \Delta) - f(x)$. For small perturbations, we linearize at $x$:

$$f(x + \Delta) = f(x) + J_f(x)\Delta + r(x, \Delta), \qquad \|r(x, \Delta)\|_2 \leq \tfrac{L(x)}{2}\|\Delta\|_2^2, \tag{1}$$

where $J_f(x) \in \mathbb{R}^{k \times d}$ is the Jacobian and $L(x)$ bounds the local Hessian.

**Gaussian as a least-favorable perturbation.** We show that, under a natural variance constraint, Gaussian noise maximizes the expected feature-space mean-squared error.

**Theorem 1** (Gaussian envelope under a variance constraint). *Let $\Delta$ be any zero-mean perturbation with covariance $\Sigma_\Delta \succeq 0$ satisfying $\Sigma_\Delta \preceq \sigma^2 I_d$. Then, under the local model Eq. 1, we have*

$$\mathbb{E}\big[\|f(x + \Delta) - f(x)\|_2^2\big] = \mathrm{tr}\big(J_f(x)\Sigma_\Delta J_f(x)^\top\big) + O\big(\mathbb{E}\|\Delta\|_2^3\big) \leq \sigma^2 \|J_f(x)\|_F^2 + O\big(\mathbb{E}\|\Delta\|_2^3\big).$$

*Moreover, if $\eta \sim \mathcal{N}(0, \sigma^2 I_d)$, then $\mathbb{E}\big[\|f(x+\eta) - f(x)\|_2^2\big] = \sigma^2 \|J_f(x)\|_F^2 + O\big(\mathbb{E}\|\eta\|_2^3\big)$, so Gaussian noise saturates this upper bound on the leading term.*

**Connection to other corruptions.** Other image corruptions, including noise, blur, brightness, weather, and digital artifacts, can be implemented as locally bounded operators, such as convolutions, pixelwise affine transforms, and compression. These yield perturbations $\Delta_{\mathcal{C},s}$ whose covariance has a bounded spectral norm $\lambda_{\max}\big(\mathrm{Cov}(\Delta_{\mathcal{C},s})\big) \leq \sigma_{\mathcal{C}}(s)^2$ for some effective variance level $\sigma_{\mathcal{C}}(s)$. Applying Theorem 1 with $\sigma = \sigma_{\mathcal{C}}(s)$ shows that, to leading order, the expected feature-space distortion induced by corruption $\mathcal{C}$ at severity $s$ is upper-bounded by that induced by Gaussian noise $\eta \sim \mathcal{N}(0, \sigma_{\mathcal{C}}(s)^2 I_d)$, which is the least favorable under the same variance budget. Furthermore, averaging over a data distribution $x \sim \mathcal{D}$ gives $\mathbb{E}_{x,\Delta}\big[\|f(x + \Delta) - f(x)\|_2^2\big] \lesssim \sigma^2 \mathbb{E}_x\big[\|J_f(x)\|_F^2\big]$, so robustness to Gaussian noise probes essentially the same Jacobian-based sensitivity that governs many common corruptions. See Appendix J and Appendix K for further discussion and empirical results.

**Limitations** Our study focuses exclusively on robustness to additive Gaussian noise, which, although common in imaging pipelines, does not encompass all real-world corruptions, such as adversarial perturbations, weather effects, or sensor-specific artifacts. Also, the empirical findings are derived from pretrained models in the `timm` library and controlled experiments on specific datasets, which may represent a limitation in their generalizability to other domains like medical imaging or video processing. Future work could extend these insights to broader corruptions, architectures, and datasets. See Appendix L for results on other architectures under a controlled setup.

## 4 NOISE ATTENUATION BY LOW-PASS KERNELS

ViTs have various configurations (Dosovitskiy et al., 2021), such as the size of each patch in the patch embedding and the input image size in pixels, which we refer to as the input resolution. Even within the

Table 1: Top-1 accuracy (%) on the ImageNet-1K dataset before and after adding Gaussian noise to images. For the rank difference (RankDiff), more negative values indicate better robustness under noise. Models with large kernels and small resolutions consistently showed improved robustness.

| Pretrained Model | Size | Patch Size | Resol. | Top-1→w/ Noise | Rank→w/ Noise | RankDiff |
|---|---|---|---|---|---|---|
| vit_small_patch16_224.augreg_in1k | S | $16^2$ | $224^2$ | 78.84→59.22 | 885→547 | -338 |
| vit_small_patch16_384.augreg_in1k | S | $16^2$ | $384^2$ | 81.12→56.59 | 673→613 | -60 |
| vit_base_patch16_224.augreg_in1k | B | $16^2$ | $224^2$ | 79.15→62.21 | 862→487 | -375 |
| vit_base_patch16_384.augreg_in1k | B | $16^2$ | $384^2$ | 81.10→60.23 | 676→524 | -152 |
| vit_base_patch32_224.augreg_in1k | B | $32^2$ | $224^2$ | 74.90→58.44 | 1075→569 | -506 |
| vit_base_patch32_384.augreg_in1k | B | $32^2$ | $384^2$ | 78.75→59.65 | 893→539 | -354 |
| vit_tiny_patch16_224.augreg_in21k_ft_in1k | T | $16^2$ | $224^2$ | 75.46→40.34 | 1060→949 | -111 |
| vit_tiny_patch16_384.augreg_in21k_ft_in1k | T | $16^2$ | $384^2$ | 78.42→30.50 | 921→1078 | +157 |
| vit_small_patch16_224.augreg_in21k_ft_in1k | S | $16^2$ | $224^2$ | 81.39→62.43 | 644→479 | -165 |
| vit_small_patch16_384.augreg_in21k_ft_in1k | S | $16^2$ | $384^2$ | 83.80→62.25 | 349→484 | +135 |
| vit_small_patch32_224.augreg_in21k_ft_in1k | S | $32^2$ | $224^2$ | 76.00→57.14 | 1044→601 | -443 |
| vit_small_patch32_384.augreg_in21k_ft_in1k | S | $32^2$ | $384^2$ | 80.48→57.33 | 740→596 | -144 |
| vit_base_patch8_224.augreg_in21k_ft_in1k | B | $8^2$ | $224^2$ | 85.80→73.50 | 145→118 | -27 |
| vit_base_patch16_224.augreg_in21k_ft_in1k | B | $16^2$ | $224^2$ | 84.53→71.19 | 257→192 | -65 |
| vit_base_patch16_384.augreg_in21k_ft_in1k | B | $16^2$ | $384^2$ | 85.99→70.89 | 129→208 | +79 |
| vit_base_patch32_224.augreg_in21k_ft_in1k | B | $32^2$ | $224^2$ | 80.71→65.31 | 719→392 | -327 |
| vit_base_patch32_384.augreg_in21k_ft_in1k | B | $32^2$ | $384^2$ | 83.35→63.72 | 412→437 | +25 |
| vit_large_patch16_224.augreg_in21k_ft_in1k | L | $16^2$ | $224^2$ | 85.84→76.62 | 141→55 | -86 |
| vit_large_patch16_384.augreg_in21k_ft_in1k | L | $16^2$ | $384^2$ | 87.08→76.23 | 59→61 | +2 |
| vit_base_patch16_224.orig_in21k_ft_in1k | B | $16^2$ | $224^2$ | 81.79→60.91 | 603→513 | -90 |
| vit_base_patch16_384.orig_in21k_ft_in1k | B | $16^2$ | $384^2$ | 84.20→54.91 | 302→657 | +355 |
| vit_base_patch8_224.augreg2_in21k_ft_in1k | B | $8^2$ | $224^2$ | 86.22→76.09 | 109→67 | -42 |
| vit_base_patch16_224.augreg2_in21k_ft_in1k | B | $16^2$ | $224^2$ | 85.10→74.50 | 203→96 | -107 |
| vit_base_patch16_224.sam_in1k | B | $16^2$ | $224^2$ | 80.24→57.13 | 771→602 | -169 |
| vit_base_patch32_224.sam_in1k | B | $32^2$ | $224^2$ | 73.69→51.33 | 1101→748 | -353 |
| vit_medium_patch16_gap_256.sw_in12k_ft_in1k | M | $16^2$ | $256^2$ | 84.45→73.07 | 274→132 | -142 |
| vit_medium_patch16_gap_384.sw_in12k_ft_in1k | M | $16^2$ | $384^2$ | 85.54→73.98 | 163→106 | -57 |
| vit_so150m_patch16_reg4_gap_256.sbb_e250_in12k_ft_in1k | B+ | $16^2$ | $256^2$ | 86.68→77.54 | 81→38 | -43 |
| vit_so150m_patch16_reg4_gap_384.sbb_e250_in12k_ft_in1k | B+ | $16^2$ | $384^2$ | 87.37→77.30 | 49→44 | -5 |

same ViT architecture, various pretrained weights are available: They were trained with different recipes, the hyperparameter combinations used in training. For example, vit_base_patch16_224.augreg_in1k indicates the ViT with a model size of base, a patch size of 16 to set the size of each patch to $16 \times 16$ pixels, a resolution of $224^2$, and pretrained weights obtained using a training recipe of AugReg (Steiner et al., 2022) and a dataset of ImageNet-1K (Deng et al., 2009). Although plenty of variations in its configuration are allowed, the effect of each choice on robustness against Gaussian noise has not been clearly studied, making it difficult for practitioners to choose which one to use.

To study the effect of each architectural factor in a ViT on robustness, we performed an extensive evaluation using pretrained ViTs with various configurations. For example, by comparing vit_base_patch16_224.augreg_in1k and vit_base_patch32_224.augreg_in1k, we can study the effect of the choice of patch sizes of 16 and 32 on performance because all other conditions remained the same. In this section, we first present empirical observations from different configurations, and then we examine the corresponding properties.

## 4.1 EMPIRICAL OBSERVATION

We used the timm library, which provides 1,174 pretrained vision models. For all pretrained models, we evaluated the top-1 accuracy (%) on the standard ImageNet-1K dataset. Then we injected Gaussian noise into the images on the ImageNet-1K dataset and measured the top-1 accuracy. We used the Albumentations.GaussNoise() function (Buslaev et al., 2020) with a scale factor with a range of (0.2, 0.44), where the noise was clipped to be [0, 1] and was fixed in our evaluation. Although it is natural to observe a linear accuracy drop after applying a specific corruption (Recht et al., 2019; Hendrycks & Dietterich, 2019), a model with robustness would show a relatively smaller drop in top-1 accuracy. Motivated by this behavior, we identified robust models by observing relative ranking among the 1,174 models: When a

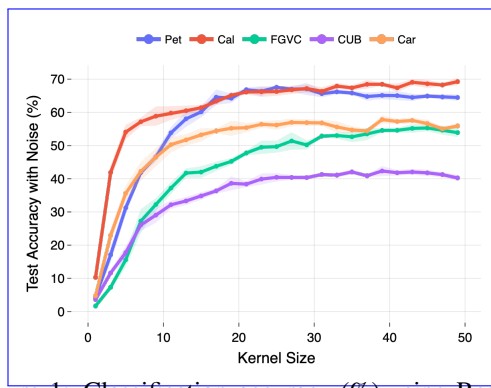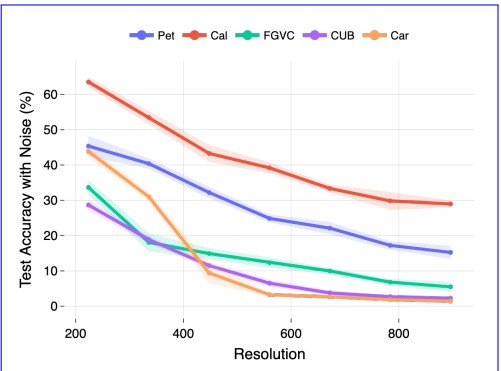

Figure 1: Classification accuracy (%) using ResNet-50 for different kernel sizes and resolutions. Larger kernels and smaller resolutions improved performance. Shaded areas represent standard deviations.

model ranked 50th becomes 20th after adding Gaussian noise, we say that it demonstrates relatively stronger robustness to Gaussian noise. To investigate the model with improved rank, we computed the rank difference before and after applying Gaussian noise, where more negative values indicate better robustness. Full rationale for rank difference and technical details are available in Appendix G and Appendix H. Full results on all models are in supplementary materials. Based on these rank differences, we compared pairs of ViTs with different configurations and investigated which architectural factors contribute to improved ranking under noise.

Table 1 summarizes the top-1 accuracy and ranking changes before and after injecting Gaussian noise. We found that the rank difference was lower when a ViT had 1) a larger patch size, such as 32, and 2) a smaller resolution, such as $224^2$. For example, comparing `vit_base_patch16_224.augreg_in1k` and `vit_base_patch32_224.augreg_in1k`, we observed that the model with a patch size of 32 yielded a lower rank difference than the one with a patch size of 16. We consistently observed similar behavior across multiple pretrained weights such as AugReg2, original ViTs, SAM, and others (Steiner et al., 2022; Chen et al., 2022). The same holds for resolution, where a model with a $224^2$ resolution exhibited a lower rank difference than one with $384^2$. Note that this observation is contrary to the common practice of scaling up resolution to improve general performance (Tan & Le, 2019); our results indicate that this practice may increase vulnerability to Gaussian noise. These two factors were significantly more important than others, such as model size.

The patch size of a ViT corresponds to the kernel size used in the patch embedding, which is referred to as the stem. Based on these observations, we investigated whether using a larger stem kernel and a smaller resolution improves robustness to Gaussian noise on another architecture, performing controlled experiments on ResNets (He et al., 2016). Specifically, we trained ResNet-50 on five datasets, including Oxford-IIIT Pet (Parkhi et al., 2012), Caltech-101 (Fei-Fei et al., 2007), FGVC-Aircraft (Maji et al., 2013), Caltech-UCSD Birds-200-2011 (Wah et al., 2011), and Stanford Cars (Krause et al., 2013) datasets. Similar to the above ViT experiments, we trained ResNet in a standard recipe (Appendix H), obtained numerous models with different kernel sizes and resolutions, and measured classification accuracy after applying Gaussian noise.

We observed that larger kernel sizes and smaller resolutions improved classification accuracy under additive Gaussian noise (Figure 1). The classification errors on noisy images tended to decrease quadratically with larger kernel sizes and smaller resolutions.

## 4.2 THEORETICAL ANALYSIS

Now, we prove that the noise energy decays quadratically with the stem kernel size and the resolution, or equivalently, the anti-aliased downsampling factor. Full proofs are available in Appendix A. Throughout,

$\eta \sim \mathcal{N}(0, \sigma^2 I)$ denotes independent and identically distributed (i.i.d.) Gaussian noise, and the per-pixel noise gain is the output noise energy normalized by the number of output pixels (Oppenheim, 1999).

**Setup.** For a kernel size $k \geq 3$, let $K_k \in \mathbb{R}^{k \times k}$ denote the linear, shift-invariant stem kernel, and $\widehat{K}_k$ its DFT (Oppenheim, 1999). We consider a single, mild assumption on the stem kernel:

- **($A_{roll}$)** Radial low-pass envelope at scale $1/k$: There exist $\beta, \delta > 0$ such that, for all frequencies $\boldsymbol{\omega}$,

$$|\widehat{K}_k(\boldsymbol{\omega})| \leq \phi_k(\|\boldsymbol{\omega}\|), \qquad \phi_k(r) := (1 + \beta k r)^{-1-\delta},$$

where $\phi_k$ is nonincreasing in $r$. This assumption works well in practical use cases (Appendix F).

**Per-pixel noise gain for stem kernel.** We define

$$\gamma(k) := \frac{\mathbb{E}\big[\|K_k * \eta\|_2^2\big]}{\sigma^2 HW} \overset{\text{(Parseval)}}{=} \frac{1}{HW} \sum_{\boldsymbol{\omega}} |\widehat{K}_k(\boldsymbol{\omega})|^2 = \|K_k\|_F^2, \tag{2}$$

where $H$ and $W$ are the height and width. Intuitively, $\gamma(k)$ is the average squared magnitude response of the stem kernel.

**Theorem 2** (Noise attenuation for practical low-pass stem kernel). *Assume* ($A_{roll}$). *Then, there exists a constant $C > 0$, independent of $k$, such that*

$$\gamma(k) = \frac{\mathbb{E}\big[\|K_k * \eta\|_2^2\big]}{\sigma^2 HW} \leq \frac{C}{k^2}.$$

*Moreover, the $k^{-2}$ rate is achievable.*

**Remark 1** (Practical reading of Theorem 2). *Doubling the stem kernel size, such as the patch size from 16 to 32, quarters the output noise energy ($\approx -6$ dB).*

**Per-output-pixel noise gain for anti-aliased downsampling.** For a downsampling factor $s \geq 1$, we define

$$D_s := (\Downarrow_s) \circ K_{g(s)}, \qquad c_1 s \leq g(s) \leq c_2 s, \tag{3}$$

*i.e.*, filter with $K_{g(s)}$ satisfying the same assumptions at scale $g(s)$ and then downsample by $s$. We normalize the noise gain by the number of output pixels:

$$\gamma_{\downarrow}(s) := \frac{\mathbb{E}\big[\|D_s \eta\|_2^2\big]}{\sigma^2 HW/s^2}. \tag{4}$$

**Theorem 3** (Resolution-driven robustness). *There exists $C' > 0$ independent of $s$ such that*

$$\gamma_{\downarrow}(s) \leq \frac{C'}{s^2}.$$

*This $s^{-2}$ rate is tight up to constants.*

**Remark 2** (Practical reading of Theorem 3). *Resizing $384^2$ to $224^2$ corresponds to $s \approx 1.71$ and yields roughly $s^{-2} \approx 0.34$ of the original noise energy per output pixel ($\approx -4.7$ dB).*

## 5 CHOICE ON POOLING

### 5.1 EMPIRICAL OBSERVATION

Extending the above analysis, we probed the effect of choosing specific architectural types of ResNets on robustness. Specifically, ResNet has several types, including ResNet-{C, D, T, S} (He et al., 2019; Wightman

Table 2: Classification accuracy (%) with different choices of ResNet type. The numbers in the parentheses represent standard deviations on the five runs with different random seeds.

| Dataset | Model | ResNet-50-T | ResNet-50-D | ResNet-50-C | ResNet-50-S |
|---------|-------|-------------|-------------|-------------|-------------|
| **Configuration** | | | | | |
| | Stem Conv | 3-layer 3×3 | 3-layer 3×3 | 3-layer 3×3 | 3-layer 3×3 |
| | Stem Width | 24, 48, 64 | 32, 32, 64 | 32, 32, 64 | 64, 64, 128 |
| | Downsample | Average | Average | Convolution | Convolution |
| **Results** | | | | | |
| Oxford-IIIT Pet | Val. Acc. w/ Noise | 39.1 (11.1) | 37.9 (9.6) | 34.9 (11.4) | 24.3 (3.3) |
| | Test Acc. w/ Noise | 38.1 (11.3) | 36.1 (10.3) | 34.0 (10.4) | 22.9 (2.4) |
| Caltech-101 | Val. Acc. w/ Noise | 62.3 (1.4) | 61.2 (3.0) | 58.8 (1.1) | 50.9 (3.2) |
| | Test Acc. w/ Noise | 59.7 (1.1) | 59.1 (2.8) | 57.8 (1.0) | 49.5 (2.7) |
| FGVC-Aircraft | Val. Acc. w/ Noise | 27.8 (1.6) | 27.3 (2.4) | 23.9 (1.9) | 4.7 (0.9) |
| | Test Acc. w/ Noise | 29.9 (1.1) | 30.4 (1.6) | 26.1 (2.1) | 5.5 (0.8) |
| Caltech-UCSD Birds-200-2011 | Val. Acc. w/ Noise | 27.6 (2.0) | 28.8 (0.8) | 26.3 (1.3) | 13.9 (0.6) |
| | Test Acc. w/ Noise | 26.3 (2.0) | 27.7 (0.6) | 25.2 (1.7) | 13.7 (1.1) |
| Stanford Cars | Val. Acc. w/ Noise | 56.9 (2.3) | 55.2 (2.8) | 41.6 (2.3) | 29.2 (1.9) |
| | Test Acc. w/ Noise | 55.0 (1.9) | 53.2 (2.7) | 40.5 (2.3) | 28.5 (2.0) |

Table 3: Classification accuracy (%) comparing different poolings. The largest gain came from AvgPool.

| Dataset | Model | MaxPool | NNPool | AvgPool |
|---------|-------|---------|--------|---------|
| Oxford-IIIT Pet | Val. Acc. w/ Noise | 42.0 (1.1) | 44.2 (2.8) | 50.2 (1.9) |
| | Test Acc. w/ Noise | 41.8 (0.9) | 42.3 (3.2) | 49.3 (1.8) |
| Caltech-101 | Val. Acc. w/ Noise | 59.5 (1.0) | 58.3 (1.1) | 62.7 (1.8) |
| | Test Acc. w/ Noise | 57.2 (1.3) | 56.7 (1.1) | 60.8 (1.9) |
| FGVC-Aircraft | Val. Acc. w/ Noise | 24.2 (3.5) | 22.8 (1.9) | 41.0 (2.9) |
| | Test Acc. w/ Noise | 27.3 (3.6) | 24.7 (1.8) | 43.1 (3.1) |
| Caltech-UCSD Birds-200-2011 | Val. Acc. w/ Noise | 26.9 (1.8) | 27.5 (3.0) | 28.8 (1.8) |
| | Test Acc. w/ Noise | 26.1 (1.7) | 25.6 (2.6) | 26.8 (1.2) |
| Stanford Cars | Val. Acc. w/ Noise | 43.3 (3.4) | 49.1 (1.9) | 52.1 (1.8) |
| | Test Acc. w/ Noise | 42.2 (2.8) | 46.9 (1.3) | 51.2 (2.1) |

et al., 2021; Guo et al., 2020), although the effects of these choices and their underlying mechanisms on robustness have been rarely studied. Here, we trained the four ResNets on the five datasets mentioned above and compared their classification accuracy after applying Gaussian noise (Table 2).

Overall, the T and D types of ResNet demonstrated robust results against Gaussian noise, followed by the C and S types of ResNet. While there are several different factors among the four ResNets (Appendix H), the core difference is the pooling in downsampling: the T and D types of ResNet adopt average pooling with convolution in downsampling, whereas the C and S types of ResNet adopt strided $1 \times 1$ convolution in downsampling, which is equivalent to nearest-neighbor pooling followed by a $1 \times 1$ convolution.

We further explored the effect of pooling choice on robustness to Gaussian noise. Using ResNet-50, we compared the original one, which uses max pooling in the stem, and modified ResNets that adopt nearest-neighbor pooling or average pooling in the stem (Table 3). ResNets with average pooling consistently

yielded robust performance against Gaussian noise among the three setups in pooling. More results for other architectures under controlled conditions are available in Appendix C.

## 5.2 THEORETICAL ANALYSIS

We explain why average pooling is more robust than max pooling under i.i.d. additive Gaussian noise.

**Setup.** Consider a pooling window of size $k \geq 2$ in a single channel. Let the clean activations be $S = (S_1, \ldots, S_k) \in \mathbb{R}^k$ and the observation be $S + \eta$ with i.i.d. noise $\eta \sim \mathcal{N}(0, \sigma^2 I_k)$. We define

$$X_{\text{avg}} := \tfrac{1}{k} \sum_{i=1}^{k} (S_i + \eta_i), \qquad X_{\text{max}} := \max_{1 \leq i \leq k} (S_i + \eta_i),$$

their clean counterparts $S_{\text{avg}} := \frac{1}{k} \sum_i S_i$, $S_{\text{max}} := \max_i S_i$, and the errors $\delta_{\text{avg}} := X_{\text{avg}} - S_{\text{avg}}$, $\delta_{\text{max}} := X_{\text{max}} - S_{\text{max}}$. Let $\Delta := S_{(1)} - S_{(2)} \geq 0$ be the gap between the largest and second-largest entries. We also denote $T_{\text{avg}}(v) := \frac{1}{k} \sum_{i=1}^{k} v_i$, $T_{\text{max}}(v) := \max_{1 \leq i \leq k} v_i$, and $\|T\|_{\ell_2 \to \ell_2}$ for $\ell_2$-Lipschitz constant.

**Theorem 4** (Average and max poolings under Gaussian noise). *For any $S \in \mathbb{R}^k$ and $\sigma > 0$, we have*

*(i)* ***Average pooling*** *is unbiased and reduces variance proportionally to the window area:* $\mathbb{E}[\delta_{\text{avg}}] = 0$, $\text{Var}[\delta_{\text{avg}}] = \sigma^2/k$.

*(ii)* ***Max pooling*** *incurs a positive noise bias and admits the following mean-squared error (MSE) controls:*

$$\text{(Bias)} \quad \mathbb{E}[\delta_{\text{max}}] = \mathbb{E}[\max_i (S_i + \eta_i)] - \max_i S_i \geq 0,$$

$$\text{(Uniform-signal case)} \quad (S_1 = \cdots = S_k): \; \delta_{\text{max}} = \sigma M_k, \; \mathbb{E}[\delta_{\text{max}}^2] = \sigma^2 \mathbb{E}[M_k^2],$$

$$\text{(General case)} \quad |\delta_{\text{max}}| \leq \|\eta\|_\infty \Rightarrow \mathbb{E}[\delta_{\text{max}}^2] \leq \sigma^2 \mathbb{E}[A_k^2],$$

*where $M_k := \max_{1 \leq i \leq k} Z_i$ and $A_k := \max_{1 \leq i \leq k} |Z_i|$ with $Z_i \overset{\text{i.i.d.}}{\sim} \mathcal{N}(0, 1)$. In particular, $\mathbb{E}[A_k^2] \leq 2\log(2k) + 2$, so $\mathbb{E}[\delta_{\text{max}}^2] \leq \sigma^2 (2\log(2k) + 2)$.*

*(iii)* ***Adversarial worst-case sensitivity.*** *For any perturbation $n \in \mathbb{R}^k$, $|\frac{1}{k} \sum_i n_i| \leq \|n\|_2/\sqrt{k}$, so $\|T_{\text{avg}}\|_{\ell_2 \to \ell_2} = k^{-1/2}$; and $|\max_i a_i - \max_i b_i| \leq \|a - b\|_\infty \leq \|a - b\|_2$, so $\|T_{\text{max}}\|_{\ell_2 \to \ell_2} \leq 1$.*

*(iv)* ***Large-gap regime.*** *For $z := \Delta/\sigma$, one has $\lim_{z \to \infty} \mathbb{E}[\delta_{\text{max}}^2] = \sigma^2$; when the top index never switches under noise, max pooling is equivalent to reading a single noisy entry.*

**Remark 3** (Practical reading of Theorem 4). *Average pooling is unbiased and cuts Gaussian noise variance by a factor $k$ (e.g., a $2 \times 2$ window gives $-6$ dB). Max pooling is positively biased, and its MSE grows at most logarithmically with the window size, while also having a greater worst-case Lipschitz gain, clearly worse than average pooling.*

**Remark 4** (Average and nearest-neighbor poolings). *Selecting a fixed element in the window, such as the nearest-neighbor pooling, is unbiased with an MSE $\sigma^2$. Hence, average pooling is strictly more robust to additive Gaussian noise than nearest-neighbor pooling by a factor of $k$ in MSE.*

## 6 WHY ARE CLIP MODELS VULNERABLE?

### 6.1 EMPIRICAL OBSERVATION

Although the original ViT (Dosovitskiy et al., 2021) was trained with supervised learning, the CLIP study (Radford et al., 2021) trained ViTs with contrastive learning and successfully achieved competitive per-

Table 4: ImageNet-1K results for ViT-B/16 $224^2$ with eight different pretrained weights. CLIP ViTs tended to yield worse ranks under noise.

| Pretrained Model | Mean-Std | Top-1 → w/ Noise | Rank → w/ Noise | RankDiff |
|---|---|---|---|---|
| `vit_base_patch16_224.augreg_in1k` | INCEPTION | 79.15 → 62.21 | 862 → 487 | -375 |
| `vit_base_patch16_224.augreg2_in21k_ft_in1k` | INCEPTION | 85.10 → 74.50 | 203 → 96 | -107 |
| `vit_base_patch16_224.orig_in21k_ft_in1k` | INCEPTION | 81.79 → 60.91 | 603 → 513 | -90 |
| `vit_base_patch16_224.augreg_in21k_ft_in1k` | INCEPTION | 84.53 → 71.19 | 257 → 192 | -65 |
| `vit_base_patch16_clip_224.openai_ft_in12k_in1k` | OPENAI | 85.94 → 70.81 | 135 → 209 | +74 |
| `vit_base_patch16_clip_224.laion2b_ft_in12k_in1k` | OPENAI | 86.17 → 71.24 | 114 → 189 | +75 |
| `vit_base_patch16_clip_224.laion2b_ft_in1k` | OPENAI | 85.47 → 67.88 | 168 → 311 | +143 |
| `vit_base_patch16_clip_224.openai_ft_in1k` | OPENAI | 85.29 → 67.06 | 182 → 340 | +158 |

formance. Currently, available pretrained weights for ViTs are largely divided into CLIP ViTs and others trained with supervised learning; we refer to the latter as supervised ViTs. The training methods and datasets differ between these two sources of ViTs, yielding different pretrained weights, while they have almost the same architecture with only a single minor difference. Nevertheless, we observed that CLIP ViTs exhibited significant vulnerabilities when Gaussian noise was applied to images (Table 4). Similar observations regarding the degraded performance of CLIP ViTs due to distribution shifts have been reported in certain studies (Shu et al., 2023; Wortsman et al., 2022); they focused on the characteristics of CLIP pretrained weights due to different datasets or training schemes, but we present a different perspective on this issue.

We performed ablation studies to identify what determined the difference in robustness (Appendix E). We discovered that the core factor in different robustness arose from the preprocessing pipeline. Specifically, CLIP ViTs apply mean-std normalization to input images using certain per-channel mean and standard deviation (std) constants, which we refer to as the OPENAI constants (Appendix H), whereas supervised ViTs apply different per-channel mean-std constants, which are often called INCEPTION constants (Szegedy et al., 2016). In other words, the OPENAI mean-std constants led to vulnerability to Gaussian noise, whereas the INCEPTION mean-std constants did not show this vulnerability.

Indeed, when we replaced the OPENAI mean-std constants with the INCEPTION constants, the CLIP ViTs achieved improved robustness (Table 5). The reverse also holds, and similar vulnerability was observed when adopting IMAGENET mean-std constants for ViTs. Full results on other datasets are available in Appendix C, where we observed these improvements across various pretrained weights with different training recipes.

### 6.2 THEORETICAL ANALYSIS

We give an explanation for the empirical vulnerability of CLIP ViTs to additive Gaussian noise. The key point is that channel-wise normalization sets the pixel-space sensitivity scale: Smaller per-channel stds in the input normalization enlarge the worst-case response to perturbations even before the backbone acts.

**Setup.** Let $x \in [0,1]^{C \times H \times W}$ be an image and $\eta$ an additive perturbation. Let $\mu \in \mathbb{R}^C$ and $\boldsymbol{\sigma} \in \mathbb{R}_{>0}^C$ be the per-channel means and stds, and define the normalization $N_{\mu,\boldsymbol{\sigma}}(x) := (x - \mu)/\boldsymbol{\sigma}$. Let $f : \mathbb{R}^{C \times H \times W} \to \mathbb{R}^K$ denote the vision backbone operating on normalized inputs, which is globally $\ell_2$-Lipschitz with constant $L_z$ on its domain.[1] We study the end-to-end pipeline $F_{\mu,\boldsymbol{\sigma}} := f \circ N_{\mu,\boldsymbol{\sigma}}$ and its $\ell_2$-Lipschitz constant $\|F_{\mu,\boldsymbol{\sigma}}\|_{\mathrm{Lip}}$.

**Theorem 5** (Pixel-space Lipschitz bound). *For any image $x$ and perturbation $\eta$, we obtain*

$$\left\|F_{\mu,\boldsymbol{\sigma}}(x+\eta) - F_{\mu,\boldsymbol{\sigma}}(x)\right\|_2 \leq L_z \left\|\frac{\eta}{\boldsymbol{\sigma}}\right\|_2 \leq \frac{L_z}{\sigma_{\min}} \|\eta\|_2,$$

*where $\sigma_{\min} := \min_c \boldsymbol{\sigma}_c$. In particular, the pixel-space Lipschitz constant satisfies $\|F_{\mu,\boldsymbol{\sigma}}\|_{\mathrm{Lip}} \leq L_z/\sigma_{\min}$.*

---

[1] This assumption holds when linear layers have bounded spectral norms and other modules are Lipschitz. ReLU: 1-Lipschitz (Gouk et al., 2021); GELU: $\approx 1.13$ (Hendrycks & Gimpel, 2016); LayerNorm: Lipschitz with a constant set by $\gamma$ and $\varepsilon$ (Ba et al., 2016).

Table 5: Classification accuracy (%) for fine-tuning ViTs on the Oxford-IIIT Pet.

| Pretrained Model | Mean-Std | Val. Acc. w/ Noise | Test Acc. w/ Noise |
|---|---|---|---|
| vit_base_patch16_clip_224.openai_ft_in12k_in1k | OPENAI | $94.5\,(1.0) \rightarrow 77.7\,(3.4)$ | $93.8\,(1.0) \rightarrow 76.3\,(4.1)$ |
| vit_base_patch16_clip_224.openai_ft_in12k_in1k | INCEPTION | $95.5\,(0.5) \rightarrow 87.3\,(2.1)$ | $95.2\,(0.6) \rightarrow 87.2\,(2.2)$ |
| vit_base_patch16_clip_224.openai_ft_in12k_in1k | IMAGENET | $94.2\,(0.4) \rightarrow 73.9\,(2.3)$ | $93.4\,(0.5) \rightarrow 72.7\,(2.5)$ |
| vit_base_patch16_clip_224.datacompxl | OPENAI | $93.6\,(0.9) \rightarrow 67.4\,(6.0)$ | $93.2\,(0.9) \rightarrow 67.3\,(5.9)$ |
| vit_base_patch16_clip_224.datacompxl | INCEPTION | $94.7\,(0.5) \rightarrow 78.5\,(4.0)$ | $93.6\,(0.6) \rightarrow 78.4\,(3.8)$ |
| vit_base_patch16_clip_224.datacompxl | IMAGENET | $92.8\,(0.9) \rightarrow 57.6\,(7.4)$ | $92.6\,(0.5) \rightarrow 58.1\,(7.4)$ |
| vit_base_patch16_clip_224.dfn2b | OPENAI | $95.0\,(0.3) \rightarrow 73.1\,(1.5)$ | $94.1\,(0.5) \rightarrow 73.3\,(1.9)$ |
| vit_base_patch16_clip_224.dfn2b | INCEPTION | $94.8\,(0.8) \rightarrow 78.6\,(4.9)$ | $93.6\,(0.4) \rightarrow 79.8\,(5.0)$ |
| vit_base_patch16_clip_224.dfn2b | IMAGENET | $95.1\,(0.3) \rightarrow 69.8\,(2.7)$ | $94.0\,(0.4) \rightarrow 68.8\,(3.1)$ |
| vit_base_patch16_clip_224.metaclip_2pt5b | OPENAI | $92.8\,(0.7) \rightarrow 64.8\,(4.4)$ | $92.0\,(0.7) \rightarrow 62.3\,(3.9)$ |
| vit_base_patch16_clip_224.metaclip_2pt5b | INCEPTION | $94.7\,(0.4) \rightarrow 78.5\,(2.0)$ | $93.9\,(0.3) \rightarrow 78.5\,(1.8)$ |
| vit_base_patch16_clip_224.metaclip_2pt5b | IMAGENET | $91.6\,(0.3) \rightarrow 54.5\,(2.5)$ | $90.8\,(0.3) \rightarrow 52.8\,(1.6)$ |
| vit_base_patch16_clip_224.openai | OPENAI | $92.5\,(0.3) \rightarrow 71.7\,(1.0)$ | $91.9\,(0.6) \rightarrow 70.2\,(1.2)$ |
| vit_base_patch16_clip_224.openai | INCEPTION | $94.0\,(0.7) \rightarrow 78.6\,(4.6)$ | $93.2\,(0.9) \rightarrow 77.3\,(5.1)$ |
| vit_base_patch16_clip_224.openai | IMAGENET | $91.2\,(0.5) \rightarrow 58.5\,(4.0)$ | $90.7\,(0.8) \rightarrow 58.4\,(4.3)$ |
| vit_base_patch16_clip_224.laion2b | OPENAI | $91.8\,(1.2) \rightarrow 56.1\,(7.7)$ | $90.5\,(1.1) \rightarrow 54.0\,(6.6)$ |
| vit_base_patch16_clip_224.laion2b | INCEPTION | $93.8\,(0.6) \rightarrow 76.4\,(1.9)$ | $92.8\,(0.5) \rightarrow 75.6\,(1.8)$ |
| vit_base_patch16_clip_224.laion2b | IMAGENET | $90.2\,(0.8) \rightarrow 52.3\,(4.4)$ | $89.5\,(0.8) \rightarrow 51.4\,(4.1)$ |
| vit_base_patch16_224.augreg_in1k | OPENAI | $95.5\,(0.2) \rightarrow 88.7\,(0.3)$ | $94.9\,(0.2) \rightarrow 88.2\,(0.7)$ |
| vit_base_patch16_224.augreg_in1k | INCEPTION | $95.5\,(0.1) \rightarrow 89.7\,(0.5)$ | $94.4\,(0.3) \rightarrow 89.2\,(0.8)$ |
| vit_base_patch16_224.augreg_in1k | IMAGENET | $95.5\,(0.2) \rightarrow 87.7\,(0.5)$ | $94.9\,(0.2) \rightarrow 87.9\,(0.7)$ |
| vit_base_patch16_224.augreg_in21k | OPENAI | $95.6\,(0.3) \rightarrow 91.4\,(0.3)$ | $95.2\,(0.5) \rightarrow 91.9\,(0.6)$ |
| vit_base_patch16_224.augreg_in21k | INCEPTION | $95.9\,(0.2) \rightarrow 92.3\,(0.3)$ | $95.6\,(0.4) \rightarrow 92.6\,(0.4)$ |
| vit_base_patch16_224.augreg_in21k | IMAGENET | $95.7\,(0.5) \rightarrow 91.6\,(0.5)$ | $95.6\,(0.3) \rightarrow 92.0\,(0.5)$ |
| vit_base_patch16_224.mae | OPENAI | $93.5\,(0.3) \rightarrow 70.8\,(2.8)$ | $93.4\,(0.2) \rightarrow 72.7\,(2.3)$ |
| vit_base_patch16_224.mae | INCEPTION | $93.7\,(0.3) \rightarrow 75.0\,(2.1)$ | $93.3\,(0.2) \rightarrow 75.2\,(2.5)$ |
| vit_base_patch16_224.mae | IMAGENET | $93.5\,(0.3) \rightarrow 72.0\,(2.0)$ | $92.7\,(0.5) \rightarrow 71.9\,(2.2)$ |

*Proof.* Write $z = N_{\mu,\boldsymbol{\sigma}}(x)$ and $\tilde{z} = N_{\mu,\boldsymbol{\sigma}}(x + \eta) = z + \eta/\boldsymbol{\sigma}$. By Lipschitzness of $f$, we have $\|f(\tilde{z}) - f(z)\|_2 \leq L_z \|\eta/\boldsymbol{\sigma}\|_2 \leq (L_z/\sigma_{\min})\|\eta\|_2$. $\qquad\square$

**Remark 5** (Practical reading of Theorem 5). *For the standard choices*

$$\boldsymbol{\sigma}_{INCEPTION} = (0.5, 0.5, 0.5), \qquad \boldsymbol{\sigma}_{CLIP} = (0.26862954, 0.26130258, 0.27577711),$$

*the worst-case pixel-space sensitivity bound for CLIP is greater by a factor*

$$\frac{L_z / \min(\boldsymbol{\sigma}_{CLIP})}{L_z / \min(\boldsymbol{\sigma}_{INCEPTION})} = \frac{0.5}{0.26130258} \approx 1.91,$$

*relative to a supervised ViT using* INCEPTION *statistics. This* $\sim 1.91\times$ *looser bound amplifies the effect of input perturbations before the feature extractor.*

## 7 CONCLUSION

Across timm models and controlled experiments, four design patterns consistently improved robustness against Gaussian noise: (1) larger stem kernel sizes, (2) smaller resolutions, (3) average pooling instead of max pooling, and (4) supervised ViTs rather than CLIP ViTs. Practically, we recommend models with these design patterns such as vit_base_patch32_224.augreg_in21k_ft_in1k for ViT-B as an example. Our analysis integrates these findings: Theorem 2 proves that noise attenuation is quadratic with stem kernel size; Theorem 3 yields an analogous gain under anti-aliased downsampling; Theorem 4 shows that average pooling is unbiased with error that falls as the window grows, whereas max pooling is positively biased and, for a uniform signal, its error grows logarithmically; and Theorem 5 explains CLIP sensitivity using pixel-space Lipschitz bounds scaling as $1/\sigma_{\min}$, which leads to a $\sim 1.91\times$ difference when comparing the OPENAI and INCEPTION constants. These insights provide actionable guidelines for practitioners to enhance the robustness of vision models against Gaussian noise in diverse applications.

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

APPENDIX TABLE OF CONTENTS

## A   PROOFS FOR THEOREMS 2 AND 3

Here, we provide proofs of the quadratic noise-decay results in Section 4.2.

### A.1   CONVENTIONS AND ASSUMPTIONS

**DFT convention and Parseval.**   For $u \in \mathbb{R}^{H \times W}$ with discrete Fourier transform (DFT) $\widehat{u}$ on the frequency grid $\Omega$, we use the Parseval identity

$$\frac{1}{HW} \sum_{\boldsymbol{\omega} \in \Omega} \left| \widehat{u}(\boldsymbol{\omega}) \right|^2 = \sum_{p \in \{1,...,H\} \times \{1,...,W\}} |u(p)|^2. \tag{5}$$

We write $\varepsilon := 2\pi / \max\{H, W\}$ for the infrared cutoff.

**Filter family.** For $k \geq 3$, let $K_k \in \mathbb{R}^{k \times k}$ denote the linear, shift-invariant stem kernel with DFT $\widehat{K}_k$. We assume only the following low-pass envelope; the same assumption applies to $K_{g(s)}$ when used as the anti-aliasing filter at scale $g(s)$:

- **($\mathbf{A_{roll}}$)** (Radial low-pass envelope at scale $1/k$) There exist $\beta, \delta > 0$ such that, for all frequencies $\boldsymbol{\omega}$,

$$|\widehat{K}_k(\boldsymbol{\omega})| \leq \phi_k(\|\boldsymbol{\omega}\|), \qquad \phi_k(r) \coloneqq (1 + \beta k r)^{-1-\delta},$$

  where $\phi_k$ is nonincreasing in $r$.

This assumption provides a monotone radial upper envelope sufficient for establishing our upper bounds: When estimating $\frac{1}{HW} \sum_{\boldsymbol{\omega}} |\widehat{K}_k(\boldsymbol{\omega})|^2$, we first dominate $|\widehat{K}_k|^2$ by $\phi_k^2$ and then apply the sum-integral comparison in Eq. 9.

**Noise model and gains.** Let $\eta \sim \mathcal{N}(0, \sigma^2 I)$ be spatially white Gaussian noise. The per-pixel noise gain of the stem kernel is

$$\gamma(k) \coloneqq \frac{\mathbb{E}\big[\|K_k * \eta\|_2^2\big]}{\sigma^2 HW} \overset{Eq.\ 5}{=} \frac{1}{HW} \sum_{\boldsymbol{\omega}} \big|\widehat{K}_k(\boldsymbol{\omega})\big|^2 = \|K_k\|_F^2. \tag{6}$$

For anti-aliased downsampling with a factor $s \geq 1$, we define

$$D_s \coloneqq (\Downarrow_s) \circ K_{g(s)}, \qquad c_1 s \leq g(s) \leq c_2 s, \tag{7}$$

and its per-output-pixel noise gain

$$\gamma_\downarrow(s) \coloneqq \frac{\mathbb{E}\big[\|D_s \eta\|_2^2\big]}{\sigma^2 HW/s^2}. \tag{8}$$

**Radial sum-integral comparison.** Let $\Omega$ be the $H \times W$ DFT grid with spacing $\varepsilon$, and let $g : [\varepsilon, \pi] \to \mathbb{R}_{\geq 0}$ be radially nonincreasing. We partition $\Omega$ into annuli $\mathcal{A}_j \coloneqq \{\boldsymbol{\omega} : j\varepsilon \leq \|\boldsymbol{\omega}\| < (j+1)\varepsilon\}$. Because each grid point occupies an area $\asymp \varepsilon^2$ and the annulus area is $2\pi r \varepsilon$ up to boundary effects, there exist absolute lattice constants $c_1, c_2 > 0$—independent of $H, W, k, s$—with

$$c_1 HW(2\pi j\varepsilon)\varepsilon \leq |\mathcal{A}_j| \leq c_2 HW(2\pi(j+1)\varepsilon)\varepsilon.$$

By monotonicity, $g((j+1)\varepsilon)|\mathcal{A}_j| \leq \sum_{\boldsymbol{\omega} \in \mathcal{A}_j} g(\|\boldsymbol{\omega}\|) \leq g(j\varepsilon)|\mathcal{A}_j|$. Summing over $j$ and dividing by $HW$ turns the lattice sum into upper and lower Riemann sums for $r \mapsto 2\pi r g(r)$ with mesh $\varepsilon$, yielding absolute constants $A_1, A_2 > 0$ such that

$$A_1 \int_\varepsilon^\pi r g(r) dr \leq \frac{1}{HW} \sum_{\boldsymbol{\omega} \in \Omega} g(\|\boldsymbol{\omega}\|) \leq A_2 \int_\varepsilon^\pi r g(r) dr. \tag{9}$$

As $\varepsilon \to 0$, both bounds converge to the same limit; for finite grids, $A_1, A_2$ absorb edge discrepancies and remain independent of the kernel scale $k$ or downsampling factor $s$.

## A.2 PROOF OF THEOREM 2 (QUADRATIC DECAY IN STEM KERNEL SIZE)

*Proof.* By Eq. 6, Eq. 9, and ($\mathbf{A_{roll}}$), we have

$$\gamma(k) \lesssim \int_\varepsilon^\pi r\big|\widehat{K}_k(r)\big|^2 dr \leq \int_\varepsilon^\pi r\big(1 + \beta k r\big)^{-2-2\delta} dr.$$

Let $u = 1 + \beta k r$. Then $r = (u-1)/(\beta k)$ and $dr = du/(\beta k)$, so

$$\int_\varepsilon^\pi r(1 + \beta k r)^{-2-2\delta} dr = \frac{1}{\beta^2 k^2} \int_{1+\beta k\varepsilon}^{1+\beta k\pi} \frac{u-1}{u^{2+2\delta}} du \leq \frac{1}{\beta^2 k^2} \int_1^\infty \frac{u-1}{u^{2+2\delta}} du = \frac{C}{k^2},$$

for a finite constant $C = C(\beta, \delta)$. Hence $\gamma(k) \leq C'/k^2$ for some $C'$ independent of $k$. $\qquad\square$

### A.3 PROOF OF THEOREM 3 (QUADRATIC DECAY UNDER ANTI-ALIASED DOWNSAMPLING)

We first state the following identity for white noise.

**Lemma 1** (Per-output-pixel gain identity). *For $D_s$ defined in Eq. 3 and white noise $\eta \sim \mathcal{N}(0, \sigma^2 I)$,*

$$\gamma_\downarrow(s) = \|K_{g(s)}\|_F^2.$$

*Proof.* Stationarity of white noise and Eq. 5 give

$$\mathbb{E}\big[\|K_{g(s)} * \eta\|_2^2\big] = (HW)\sigma^2 \|K_{g(s)}\|_F^2.$$

Downsampling by $s$ keeps every $s$-th sample along each axis: The retained samples all have equal variance as the original, pre-downsampled field. Therefore,

$$\mathbb{E}\big[\|D_s \eta\|_2^2\big] = \frac{HW}{s^2} \sigma^2 \|K_{g(s)}\|_F^2,$$

and the normalization in Eq. 8 yields $\gamma_\downarrow(s) = \|K_{g(s)}\|_F^2$. $\qquad\square$

*Proof of Theorem 3.* By Lemma 1, $\gamma_\downarrow(s) = \|K_{g(s)}\|_F^2$. Applying Theorem 2 with kernel size $k = g(s)$ gives

$$\gamma_\downarrow(s) \le \frac{C}{g(s)^2} \le \frac{C}{(c_1 s)^2} = \frac{C'}{s^2},$$

with $C' = C/c_1^2$ independent of $s$. $\qquad\square$

## B PROOF OF THEOREM 4 (AVERAGE AND MAX POOLINGS UNDER GAUSSIAN NOISE)

Consider a pooling window of size $k \ge 2$ in a single channel. Let the clean activations be $S = (S_1, \dots, S_k) \in \mathbb{R}^k$ and let the observation be $S + \eta$, where $\eta = (\eta_1, \dots, \eta_k) \overset{\text{i.i.d.}}{\sim} \mathcal{N}(0, \sigma^2)$. We define

$$X_{\text{avg}} := \frac{1}{k} \sum_{i=1}^k (S_i + \eta_i), \qquad X_{\max} := \max_{1 \le i \le k} (S_i + \eta_i),$$

and their clean counterparts $S_{\text{avg}} = \frac{1}{k} \sum_i S_i$, $S_{\max} = \max_i S_i$. Let the errors be $\delta_{\text{avg}} := X_{\text{avg}} - S_{\text{avg}}$, $\delta_{\max} := X_{\max} - S_{\max}$. Write the order statistics $S_{(1)} \ge \cdots \ge S_{(k)}$, define the gap $\Delta := S_{(1)} - S_{(2)} \ge 0$, and the standardized gap $z := \Delta/\sigma$. We use $Z_i \overset{\text{i.i.d.}}{\sim} \mathcal{N}(0, 1)$, $M_k := \max_{1 \le i \le k} Z_i$, and $A_k := \max_{1 \le i \le k} |Z_i|$.

*Proof of* (i). By definition, $\delta_{\text{avg}} = \frac{1}{k} \sum_{i=1}^k \eta_i$. Hence

$$\mathbb{E}[\delta_{\text{avg}}] = \frac{1}{k} \sum_i \mathbb{E}[\eta_i] = 0, \qquad \text{Var}[\delta_{\text{avg}}] = \frac{1}{k^2} \sum_i \text{Var}[\eta_i] = \frac{\sigma^2}{k}.$$

This part requires only i.i.d. zero-mean noise with variance $\sigma^2$. $\qquad\square$

*Proof of* (ii). (Positive bias) Let $i^\star \in \arg\max_i S_i$. Then $X_{\max} \ge S_{i^\star} + \eta_{i^\star}$. Taking expectations and using $\mathbb{E}[\eta_{i^\star}] = 0$ yields

$$\mathbb{E}[\delta_{\max}] = \mathbb{E}[X_{\max} - S_{\max}] \ge 0.$$

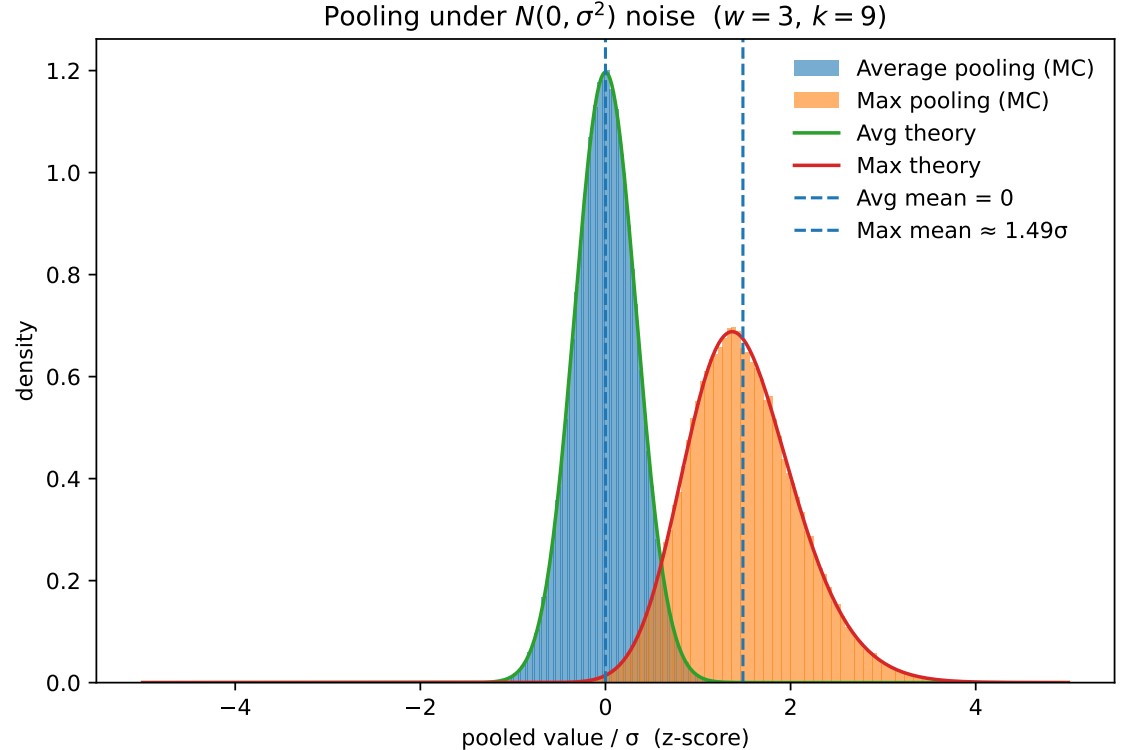

Figure 2: Illustration of positive bias introduced by max pooling

(Uniform-signal case) If $S_1 = \cdots = S_k$, translate so $S_i \equiv 0$. Then $\delta_{\max} = \max_i \eta_i = \sigma M_k$ and

$$\mathbb{E}[\delta_{\max}^2] = \sigma^2 \mathbb{E}[M_k^2].$$

Classical Gaussian extreme-value asymptotics (Hall, 1979) give

$$\mathbb{E}[M_k] = \sqrt{2\log k} - \frac{\log\log k + \log(4\pi)}{2\sqrt{2\log k}} + o\left((\log k)^{-1/2}\right), \quad \mathrm{Var}[M_k] = \frac{\pi^2}{12\log k} + o\left((\log k)^{-1}\right),$$

hence

$$\mathbb{E}[M_k^2] = \mathrm{Var}[M_k] + \left(\mathbb{E}[M_k]\right)^2 = 2\log k - \log\log k - \log(4\pi) + o(1),$$

Because $\delta_{\max} = \sigma M_k$, we have

$$\mathbb{E}[\delta_{\max}^2] = \sigma^2[\mathbb{E}M_k^2] = \sigma^2(2\log k - \log\log k - \log(4\pi) + o(1)) = \Theta(\sigma^2 \log k),$$

so the MSE scales as $\Theta(\sigma^2 \log k)$.

(General case) For any realization,

$$\left|\delta_{\max}\right| = \left|\max_i(S_i + \eta_i) - \max_i S_i\right| \le \max_i |\eta_i| = \sigma A_k.$$

Hence

$$\mathbb{E}[\delta_{\max}^2] \le \sigma^2 \mathbb{E}[A_k^2].$$

We now bound $\mathbb{E}[A_k^2]$ explicitly.

**Lemma 2.** *For* $A_k = \max_{1 \leq i \leq k} |Z_i|$ *with* $Z_i \overset{i.i.d.}{\sim} \mathcal{N}(0,1)$, *we have* $\mathbb{E}[A_k^2] \leq 2 \log(2k) + 2$.

*Proof of Lemma 2.* For $t \geq 0$, $\Pr(A_k \geq t) \leq \sum_{i=1}^{k} \Pr(|Z_i| \geq t) \leq 2ke^{-t^2/2}$, where the last step uses the union bound and the standard Gaussian tail estimate $\Pr(|Z| \geq t) \leq 2e^{-t^2/2}$ for $Z \sim \mathcal{N}(0,1)$; see, *e.g.*, Vershynin (2018). Using $\mathbb{E}[X^2] = \int_0^\infty 2t \Pr(X \geq t)dt$ for a nonnegative $X$ and splitting at $t_0 := \sqrt{2 \log(2k)}$,

$$\mathbb{E}[A_k^2] = \int_0^{t_0} 2t \Pr(A_k \geq t)dt + \int_{t_0}^\infty 2t \Pr(A_k \geq t)dt$$

$$\leq t_0^2 + \int_{t_0}^\infty 4kte^{-t^2/2}dt = 2 \log(2k) + 4ke^{-t_0^2/2}.$$

Because $e^{-t_0^2/2} = e^{-\log(2k)} = 1/(2k)$, the last term equals 2, proving the claim. $\square$

By Lemma 2,
$$\mathbb{E}[\delta_{\max}^2] \leq \sigma^2 \big(2 \log(2k) + 2\big).$$

$\square$

*Proof of* (iii). Let $T_{\text{avg}}(n) = \frac{1}{k} \sum_i n_i$. By Cauchy–Schwarz, $|T_{\text{avg}}(n)| \leq \|n\|_2 \|k^{-1}(1, \ldots, 1)\|_2 = \|n\|_2/\sqrt{k}$, so $\|T_{\text{avg}}\|_{\ell_2 \to \ell_2} = k^{-1/2}$, tight for constant $n$. For max, for any $a, b$, $|\max_i a_i - \max_i b_i| \leq \|a - b\|_\infty \leq \|a - b\|_2$, hence $\|T_{\max}\|_{\ell_2 \to \ell_2} \leq 1$, tight for a one-hot $n$. $\square$

*Proof of* (iv). Translate so $S_{(1)} = 0$ and $S_i \leq -\Delta$ for $i \geq 2$. Let $\mathcal{S}$ be the switch event that some $j \geq 2$ overtakes the top index after noise:

$$\mathcal{S} := \{\exists j \geq 2 : \eta_j - \Delta \geq \eta_1\} = \{\exists j \geq 2 : Z_j - Z_1 \geq z\}.$$

Because $Z_j$ and $Z_1$ are independent standard normals, we have $Z_j - Z_1 \sim \mathcal{N}(0, 2)$; hence, by a union bound $\Pr(\mathcal{S}) \leq (k-1) \Pr(\mathcal{N}(0,2) \geq z) \leq (k-1)e^{-z^2/4} \to 0$ as $z \to \infty$. On $\mathcal{S}^c$, $X_{\max} = S_{(1)} + \eta_1 = \eta_1$, so $\delta_{\max}^2 = \eta_1^2$. Dominated convergence then gives $\mathbb{E}[\delta_{\max}^2] \to \mathbb{E}[\eta_1^2] = \sigma^2$. $\square$

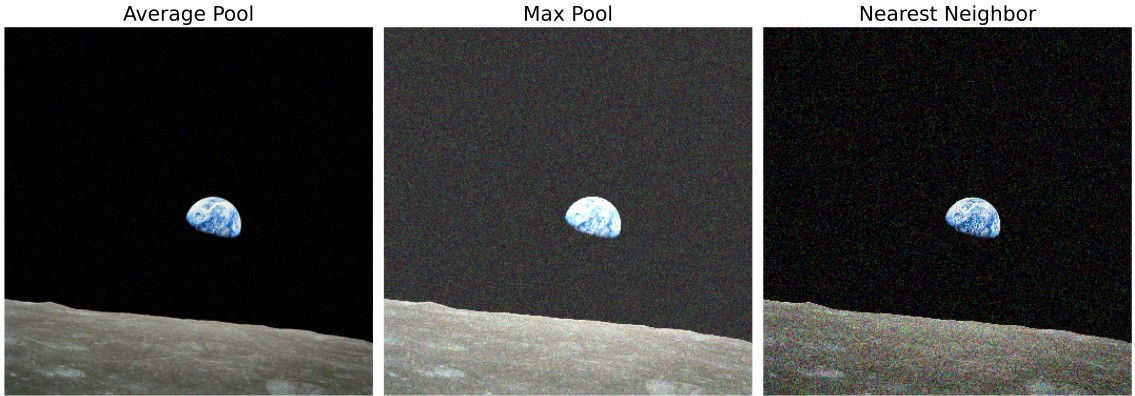

Figure 3: Examples of pooling outputs from a noisy image using average, max, and nearest neighbor

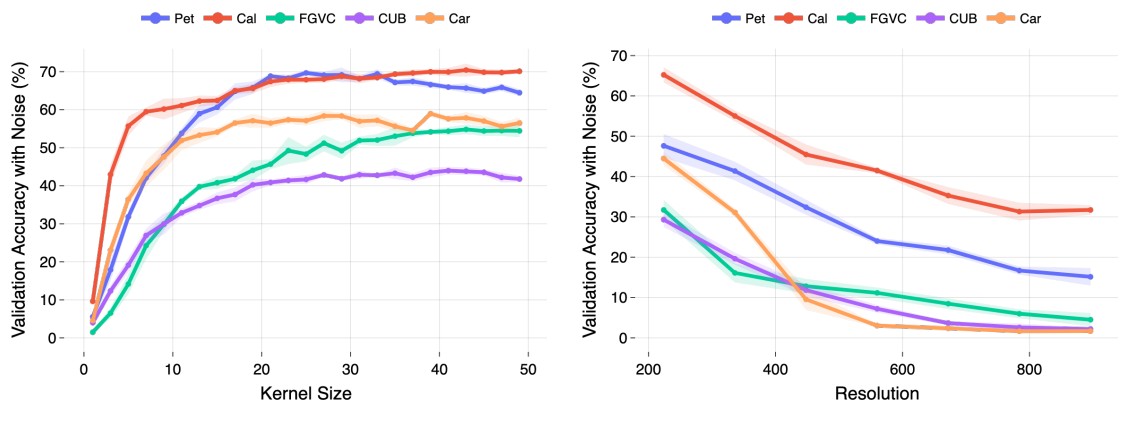

Figure 4: The results on the validation set

Table 6: Classification accuracy comparing different poolings, using ResNet-50-D

| Dataset | Model | MaxPool | NNPool | AvgPool |
|---------|-------|---------|--------|---------|
| Oxford-IIIT Pet | Val. Acc. | 87.7 (0.6) | 87.6 (0.4) | 86.7 (0.5) |
| | Test Acc. | 85.3 (0.8) | 84.7 (0.6) | 84.8 (0.9) |
| | Val. Acc. w/ Noise | 48.3 (2.2) | 46.3 (1.9) | 54.0 (3.7) |
| | Test Acc. w/ Noise | 47.8 (1.3) | 45.2 (2.3) | 53.6 (2.7) |
| Caltech-101 | Val. Acc. | 81.3 (0.7) | 82.4 (1.1) | 81.7 (0.5) |
| | Test Acc. | 80.5 (0.3) | 80.7 (0.4) | 81.6 (0.7) |
| | Val. Acc. w/ Noise | 61.1 (1.5) | 60.3 (1.4) | 62.7 (1.4) |
| | Test Acc. w/ Noise | 59.8 (1.7) | 58.3 (1.3) | 61.6 (1.3) |
| FGVC-Aircraft | Val. Acc. | 68.1 (0.2) | 67.7 (0.8) | 69.0 (0.7) |
| | Test Acc. | 68.8 (1.1) | 68.3 (1.5) | 69.6 (0.3) |
| | Val. Acc. w/ Noise | 27.7 (1.6) | 24.8 (1.8) | 42.9 (1.7) |
| | Test Acc. w/ Noise | 31.5 (2.1) | 26.9 (0.8) | 44.8 (1.1) |
| Caltech-UCSD Birds-200-2011 | Val. Acc. | 69.8 (0.7) | 69.8 (0.4) | 69.3 (1.1) |
| | Test Acc. | 67.3 (0.4) | 66.4 (0.6) | 65.9 (0.4) |
| | Val. Acc. w/ Noise | 26.8 (0.6) | 28.7 (1.7) | 31.8 (1.6) |
| | Test Acc. w/ Noise | 26.0 (0.7) | 27.4 (1.2) | 31.1 (2.1) |
| Stanford Cars | Val. Acc. | 86.5 (0.5) | 85.7 (0.5) | 84.9 (0.2) |
| | Test Acc. | 84.8 (0.2) | 83.6 (0.3) | 83.2 (0.3) |
| | Val. Acc. w/ Noise | 56.0 (0.5) | 53.6 (1.6) | 56.8 (2.2) |
| | Test Acc. w/ Noise | 54.8 (1.5) | 51.6 (1.5) | 55.3 (2.0) |

## C  ADDITIONAL EXPERIMENTAL RESULTS

Figure 4 shows the accuracy on the validation set for the controlled experiments on kernel size and resolution.

We also report additional results for ResNet-50-D (Table 6) and ResNet-101-D (Table 7) for different choices of pooling.

Table 7: Classification accuracy comparing different poolings, using ResNet-101-D

| Dataset | Model | MaxPool | NNPool | AvgPool |
|---|---|---|---|---|
| Oxford-IIIT Pet | Val. Acc. | 87.0 (0.5) | 86.5 (0.8) | 86.2 (0.3) |
| | Test Acc. | 84.8 (0.6) | 84.4 (0.7) | 84.3 (0.6) |
| | Val. Acc. w/ Noise | 52.3 (1.9) | 51.0 (1.5) | 56.4 (2.1) |
| | Test Acc. w/ Noise | 51.0 (1.3) | 49.2 (1.4) | 56.3 (2.4) |
| Caltech-101 | Val. Acc. | 82.0 (0.9) | 82.9 (0.6) | 82.9 (0.5) |
| | Test Acc. | 80.6 (0.4) | 80.7 (0.9) | 81.2 (0.4) |
| | Val. Acc. w/ Noise | 63.4 (1.9) | 63.7 (1.3) | 64.8 (1.1) |
| | Test Acc. w/ Noise | 62.1 (1.5) | 61.6 (1.7) | 63.7 (1.4) |
| FGVC-Aircraft | Val. Acc. | 69.5 (0.3) | 67.7 (0.6) | 69.4 (0.8) |
| | Test Acc. | 71.0 (1.0) | 67.1 (0.4) | 69.6 (0.7) |
| | Val. Acc. w/ Noise | 36.9 (4.0) | 28.5 (2.7) | 48.4 (1.5) |
| | Test Acc. w/ Noise | 39.1 (3.5) | 30.5 (2.6) | 49.5 (1.8) |
| Caltech-UCSD Birds-200-2011 | Val. Acc. | 70.5 (0.5) | 70.0 (0.7) | 68.9 (0.6) |
| | Test Acc. | 67.4 (0.7) | 66.8 (0.4) | 66.0 (0.7) |
| | Val. Acc. w/ Noise | 29.7 (1.7) | 29.3 (2.0) | 33.4 (1.8) |
| | Test Acc. w/ Noise | 29.0 (1.7) | 29.2 (2.6) | 32.2 (1.5) |
| Stanford Cars | Val. Acc. | 84.5 (0.4) | 83.9 (0.4) | 83.7 (0.5) |
| | Test Acc. | 83.3 (0.2) | 81.9 (0.8) | 82.1 (0.6) |
| | Val. Acc. w/ Noise | 57.5 (1.2) | 55.2 (0.9) | 58.2 (1.2) |
| | Test Acc. w/ Noise | 56.0 (0.9) | 54.5 (0.7) | 56.4 (1.2) |

Table 8 summarizes the results for ResNet-AA, which adopts anti-aliasing average pooling architecture (Zhang, 2019). Specifically, ResNet-AA adopts average pooling in all downsampling layers as well as replacing the max pooling in the stem with average pooling. ResNet-AA was marginally more robust than the ResNet with average pooling only in the stem, but not as significant as the difference with the original ResNet. The result indicates that the core difference in robustness was caused by the use of average pooling in the stem.

Table 9 summarizes ImageNet-1K results for other ViT configurations, including different patch sizes, resolutions, and training recipes.

Table 10, Table 11, Table 12, and Table 13 summarize full results for fine-tuning ViTs on other datasets. When we replaced the OPENAI mean-std constants with the INCEPTION constants, the CLIP ViTs achieved improved robustness.

## D  EXTENSION TO OTHER NOISE MODELS

We select Gaussian noise due to its approximation of aggregate perturbations by the central limit theorem and its prevalence in real-world imaging, such as sensor readout and thermal noise. Here, we explain how our main findings—noise attenuation by larger stem kernels and smaller input resolution (Theorems 2, 3), the pooling comparison (Theorem 4), and the normalization effect (Theorem 5)—extend beyond Gaussian noise.

**Setup.** We continue to use $k$ for a filter side length. For pooling windows, we use $w$ for side length and $m = w^2$ for the number of elements.

Table 8: Results on ResNet-AA

| Dataset | Model | ResNet-AA-50 | ResNet-AA-50-D | ResNet-AA-101-D |
|---|---|---|---|---|
| Oxford-IIIT Pet | Val. Acc. | 84.8 (0.8) | 86.9 (0.4) | 86.2 (0.3) |
| | Test Acc. | 83.1 (0.8) | 84.7 (0.8) | 84.3 (0.2) |
| | Val. Acc. w/ Noise | 50.1 (2.7) | 55.6 (2.0) | 58.1 (2.7) |
| | Test Acc. w/ Noise | 49.6 (3.2) | 53.9 (1.4) | 58.9 (2.9) |
| Caltech-101 | Val. Acc. | 80.2 (0.4) | 81.7 (0.7) | 83.0 (0.3) |
| | Test Acc. | 79.5 (0.6) | 80.6 (0.5) | 80.9 (0.5) |
| | Val. Acc. w/ Noise | 61.2 (1.6) | 61.7 (2.2) | 65.0 (1.5) |
| | Test Acc. w/ Noise | 60.1 (1.5) | 60.8 (2.8) | 63.3 (1.3) |
| FGVC-Aircraft | Val. Acc. | 67.3 (0.5) | 69.8 (1.0) | 69.1 (0.6) |
| | Test Acc. | 67.1 (0.9) | 70.7 (1.2) | 70.0 (0.9) |
| | Val. Acc. w/ Noise | 40.4 (3.6) | 45.5 (2.5) | 49.0 (2.9) |
| | Test Acc. w/ Noise | 42.3 (3.9) | 48.3 (2.2) | 49.5 (2.5) |
| Caltech-UCSD Birds-200-2011 | Val. Acc. | 65.3 (0.6) | 68.9 (0.8) | 69.4 (0.6) |
| | Test Acc. | 62.3 (1.1) | 66.1 (0.6) | 66.1 (0.4) |
| | Val. Acc. w/ Noise | 28.6 (0.8) | 32.5 (1.0) | 31.7 (2.8) |
| | Test Acc. w/ Noise | 27.5 (1.3) | 31.4 (1.9) | 31.0 (2.4) |
| Stanford Cars | Val. Acc. | 79.9 (0.6) | 85.9 (0.3) | 83.5 (0.6) |
| | Test Acc. | 78.9 (0.6) | 83.9 (0.4) | 81.6 (0.8) |
| | Val. Acc. w/ Noise | 51.8 (1.6) | 60.3 (2.8) | 57.2 (3.2) |
| | Test Acc. w/ Noise | 50.3 (1.0) | 58.9 (2.0) | 56.0 (3.2) |

Table 9: ImageNet-1K results for other ViT configurations

| Pretrained Model | Mean-Std | Top-1 → w/ Noise | Rank → w/ Noise | RankDiff |
|---|---|---|---|---|
| vit_base_patch16_384.augreg_in1k | INCEPTION | 81.10 → 60.23 | 676 → 524 | -152 |
| vit_base_patch16_384.augreg_in21k_ft_in1k | INCEPTION | 85.99 → 70.89 | 129 → 208 | +79 |
| vit_base_patch16_clip_384.laion2b_ft_in12k_in1k | OPENAI | 87.21 → 70.38 | 55 → 227 | +172 |
| vit_base_patch16_clip_384.openai_ft_in1k | OPENAI | 86.20 → 68.55 | 110 → 285 | +175 |
| vit_base_patch16_clip_384.openai_ft_in12k_in1k | OPENAI | 87.03 → 69.11 | 61 → 269 | +208 |
| vit_base_patch16_clip_384.laion2b_ft_in1k | OPENAI | 86.62 → 66.63 | 83 → 348 | +265 |
| vit_base_patch32_224.augreg_in1k | INCEPTION | 74.90 → 58.44 | 1075 → 569 | -506 |
| vit_base_patch32_224.sam_in1k | INCEPTION | 73.69 → 51.33 | 1101 → 748 | -353 |
| vit_base_patch32_224.augreg_in21k_ft_in1k | INCEPTION | 80.71 → 65.31 | 719 → 392 | -327 |
| vit_base_patch32_clip_224.openai_ft_in1k | OPENAI | 81.93 → 63.94 | 591 → 428 | -163 |
| vit_base_patch32_clip_224.laion2b_ft_in1k | OPENAI | 82.58 → 63.09 | 504 → 450 | -54 |
| vit_base_patch32_clip_224.laion2b_ft_in12k_in1k | OPENAI | 83.30 → 65.57 | 419 → 386 | -33 |
| vit_base_patch32_384.augreg_in1k | INCEPTION | 78.75 → 59.65 | 893 → 539 | -354 |
| vit_base_patch32_384.augreg_in21k_ft_in1k | INCEPTION | 83.35 → 63.72 | 412 → 437 | +25 |
| vit_base_patch32_clip_384.openai_ft_in12k_in1k | OPENAI | 85.21 → 68.40 | 191 → 293 | +102 |
| vit_base_patch32_clip_384.laion2b_ft_in12k_in1k | OPENAI | 85.37 → 65.58 | 180 → 383 | +203 |

**Poisson noise.** Let $S$ be the filter support with $|S| = m = k^2$. Let $h = \{h_t\}_{t \in S}$ denote the linear filter coefficients on $S$, and $Y_t \sim \text{Poisson}(x_t)$ independent. For a locally constant intensity on the filter support, where $x_t \approx \bar{x}$ in a smooth patch, we have

$$\text{Var}[\sum_t h_t Y_t] = \sum_t h_t^2 \text{Var}[Y_t] = \sum_t h_t^2 x_t = \bar{x} \sum_t h_t^2 + \sum_t h_t^2 (x_t - \bar{x}) \approx \bar{x} \|h\|_2^2, \tag{10}$$

Table 10: Classification accuracy (%) for fine-tuning ViTs on the Caltech-101.

| Pretrained Model | Mean-Std | Val. Acc. w/ Noise | Test Acc. w/ Noise |
|---|---|---|---|
| vit_base_patch16_clip_224.openai_ft_in12k_in1k | OPENAI | 93.1 (0.6) → 84.1 (1.1) | 92.0 (0.8) → 81.8 (1.4) |
| vit_base_patch16_clip_224.openai_ft_in12k_in1k | INCEPTION | 95.7 (0.6) → 90.4 (0.8) | 94.5 (0.7) → 89.5 (1.2) |
| vit_base_patch16_clip_224.openai_ft_in12k_in1k | IMAGENET | 91.6 (1.2) → 80.5 (2.4) | 90.5 (0.8) → 78.5 (2.4) |
| vit_base_patch16_clip_224.datacompxl | OPENAI | 95.3 (0.8) → 86.4 (2.3) | 94.6 (0.6) → 84.8 (2.1) |
| vit_base_patch16_clip_224.datacompxl | INCEPTION | 96.2 (0.6) → 91.0 (1.3) | 95.7 (0.9) → 89.7 (1.4) |
| vit_base_patch16_clip_224.datacompxl | IMAGENET | 94.7 (0.7) → 82.5 (1.9) | 93.8 (1.0) → 80.8 (2.6) |
| vit_base_patch16_clip_224.dfn2b | OPENAI | 90.2 (11.2) → 80.1 (15.0) | 88.9 (12.3) → 78.8 (14.8) |
| vit_base_patch16_clip_224.dfn2b | INCEPTION | 96.5 (0.6) → 91.7 (1.2) | 95.9 (0.5) → 91.0 (1.8) |
| vit_base_patch16_clip_224.dfn2b | IMAGENET | 93.7 (3.9) → 79.9 (10.5) | 92.4 (4.6) → 78.2 (10.7) |
| vit_base_patch16_clip_224.metaclip_2pt5b | OPENAI | 94.9 (0.7) → 81.5 (2.0) | 94.2 (0.7) → 79.5 (2.0) |
| vit_base_patch16_clip_224.metaclip_2pt5b | INCEPTION | 96.0 (0.5) → 89.5 (2.1) | 95.0 (0.8) → 87.8 (2.8) |
| vit_base_patch16_clip_224.metaclip_2pt5b | IMAGENET | 93.6 (1.0) → 76.3 (3.2) | 92.3 (1.2) → 74.6 (2.9) |
| vit_base_patch16_clip_224.openai | OPENAI | 92.8 (0.2) → 78.9 (3.1) | 91.7 (1.1) → 76.9 (3.6) |
| vit_base_patch16_clip_224.openai | INCEPTION | 95.4 (0.3) → 87.8 (0.9) | 95.4 (0.6) → 86.9 (0.9) |
| vit_base_patch16_clip_224.openai | IMAGENET | 92.3 (0.4) → 80.3 (1.8) | 91.8 (0.7) → 77.7 (1.9) |
| vit_base_patch16_clip_224.laion2b | OPENAI | 92.3 (0.9) → 77.2 (2.4) | 91.2 (0.6) → 75.6 (1.6) |
| vit_base_patch16_clip_224.laion2b | INCEPTION | 95.3 (0.6) → 87.3 (0.3) | 94.3 (0.6) → 85.8 (0.5) |
| vit_base_patch16_clip_224.laion2b | IMAGENET | 90.1 (0.8) → 71.5 (2.4) | 89.2 (0.5) → 67.6 (2.4) |
| vit_base_patch16_224.augreg_in1k | OPENAI | 94.4 (0.3) → 84.8 (0.9) | 94.1 (0.3) → 85.7 (0.4) |
| vit_base_patch16_224.augreg_in1k | INCEPTION | 94.1 (0.3) → 86.0 (0.5) | 93.8 (0.2) → 86.7 (0.8) |
| vit_base_patch16_224.augreg_in1k | IMAGENET | 94.3 (0.6) → 84.7 (0.6) | 94.0 (0.3) → 85.7 (0.7) |
| vit_base_patch16_224.augreg_in21k | OPENAI | 97.0 (0.4) → 95.1 (0.5) | 96.3 (0.4) → 94.5 (0.7) |
| vit_base_patch16_224.augreg_in21k | INCEPTION | 97.1 (0.3) → 95.8 (0.5) | 96.6 (0.2) → 95.4 (0.3) |
| vit_base_patch16_224.augreg_in21k | IMAGENET | 97.2 (0.2) → 95.1 (0.2) | 96.6 (0.5) → 94.6 (0.5) |
| vit_base_patch16_224.mae | OPENAI | 92.0 (0.5) → 76.3 (0.7) | 91.6 (0.8) → 75.7 (1.2) |
| vit_base_patch16_224.mae | INCEPTION | 91.6 (0.6) → 80.8 (1.2) | 91.7 (0.4) → 79.4 (0.7) |
| vit_base_patch16_224.mae | IMAGENET | 91.7 (0.5) → 75.4 (0.6) | 91.6 (0.4) → 74.5 (1.2) |

Table 11: Classification accuracy (%) for fine-tuning ViTs on the FGVC-Aircraft.

| Pretrained Model | Mean-Std | Val. Acc. w/ Noise | Test Acc. w/ Noise |
|---|---|---|---|
| vit_base_patch16_clip_224.openai_ft_in12k_in1k | OPENAI | 62.6 (1.7) → 46.6 (1.9) | 61.7 (1.2) → 47.4 (1.7) |
| vit_base_patch16_clip_224.openai_ft_in12k_in1k | INCEPTION | 60.4 (23.6) → 50.8 (20.4) | 59.5 (23.7) → 50.9 (21.1) |
| vit_base_patch16_clip_224.openai_ft_in12k_in1k | IMAGENET | 59.5 (1.4) → 44.0 (1.6) | 58.2 (1.2) → 45.2 (1.2) |
| vit_base_patch16_clip_224.datacompxl | OPENAI | 73.7 (4.9) → 50.7 (7.7) | 72.2 (4.0) → 52.7 (7.1) |
| vit_base_patch16_clip_224.datacompxl | INCEPTION | 80.8 (1.9) → 66.3 (3.7) | 79.4 (2.0) → 66.3 (3.5) |
| vit_base_patch16_clip_224.datacompxl | IMAGENET | 65.9 (4.1) → 40.0 (6.1) | 65.0 (3.4) → 41.5 (5.2) |
| vit_base_patch16_clip_224.dfn2b | OPENAI | 75.4 (4.9) → 55.7 (7.0) | 75.2 (5.2) → 57.3 (8.0) |
| vit_base_patch16_clip_224.dfn2b | INCEPTION | 82.0 (4.1) → 70.0 (8.1) | 81.7 (4.3) → 70.7 (7.6) |
| vit_base_patch16_clip_224.dfn2b | IMAGENET | 72.9 (6.6) → 51.3 (9.4) | 71.3 (7.1) → 52.5 (9.8) |
| vit_base_patch16_clip_224.metaclip_2pt5b | OPENAI | 68.0 (2.7) → 48.4 (4.1) | 67.3 (2.5) → 49.7 (3.0) |
| vit_base_patch16_clip_224.metaclip_2pt5b | INCEPTION | 80.5 (1.6) → 68.4 (3.2) | 79.2 (2.2) → 69.5 (3.4) |
| vit_base_patch16_clip_224.metaclip_2pt5b | IMAGENET | 64.5 (1.3) → 40.9 (2.4) | 64.2 (1.4) → 43.3 (2.3) |
| vit_base_patch16_clip_224.openai | OPENAI | 63.7 (4.7) → 47.4 (5.6) | 61.9 (4.3) → 49.1 (4.4) |
| vit_base_patch16_clip_224.openai | INCEPTION | 74.6 (3.5) → 65.4 (4.5) | 73.4 (3.8) → 66.0 (5.4) |
| vit_base_patch16_clip_224.openai | IMAGENET | 60.3 (1.6) → 42.6 (2.7) | 59.4 (1.4) → 43.4 (2.6) |
| vit_base_patch16_clip_224.laion2b | OPENAI | 59.9 (1.9) → 37.7 (2.4) | 58.4 (1.7) → 38.5 (1.9) |
| vit_base_patch16_clip_224.laion2b | INCEPTION | 69.2 (4.4) → 54.3 (5.5) | 68.9 (5.4) → 55.0 (6.0) |
| vit_base_patch16_clip_224.laion2b | IMAGENET | 58.3 (1.8) → 36.0 (2.3) | 56.9 (1.3) → 37.3 (2.4) |
| vit_base_patch16_224.augreg_in1k | OPENAI | 67.8 (0.8) → 50.7 (1.9) | 67.0 (1.2) → 51.2 (1.7) |
| vit_base_patch16_224.augreg_in1k | INCEPTION | 67.0 (0.5) → 52.4 (1.4) | 67.2 (0.9) → 53.6 (1.0) |
| vit_base_patch16_224.augreg_in1k | IMAGENET | 67.4 (0.4) → 50.1 (1.4) | 67.3 (0.8) → 51.0 (2.5) |
| vit_base_patch16_224.augreg_in21k | OPENAI | 78.2 (0.3) → 69.9 (0.5) | 77.2 (0.6) → 69.4 (1.1) |
| vit_base_patch16_224.augreg_in21k | INCEPTION | 78.6 (0.6) → 71.6 (0.4) | 77.3 (0.4) → 71.0 (0.4) |
| vit_base_patch16_224.augreg_in21k | IMAGENET | 77.8 (0.6) → 68.9 (0.9) | 77.1 (1.0) → 68.5 (1.2) |
| vit_base_patch16_224.mae | OPENAI | 69.3 (0.7) → 39.9 (4.2) | 68.8 (1.5) → 40.3 (4.4) |
| vit_base_patch16_224.mae | INCEPTION | 69.1 (0.7) → 43.5 (2.8) | 69.1 (0.9) → 44.0 (2.3) |
| vit_base_patch16_224.mae | IMAGENET | 69.1 (0.6) → 40.0 (2.1) | 69.4 (1.2) → 41.8 (1.2) |

Table 12: Classification accuracy (%) for fine-tuning ViTs on the Caltech-UCSD Birds-200-2011.

| Pretrained Model | Mean-Std | Val. Acc. w/ Noise | Test Acc. w/ Noise |
|---|---|---|---|
| vit_base_patch16_clip_224.openai_ft_in12k_in1k | OPENAI | 84.0 (0.9) → 64.0 (1.7) | 81.3 (1.0) → 61.1 (1.1) |
| vit_base_patch16_clip_224.openai_ft_in12k_in1k | INCEPTION | 85.3 (1.6) → 69.3 (1.7) | 82.7 (1.3) → 67.0 (2.5) |
| vit_base_patch16_clip_224.openai_ft_in12k_in1k | IMAGENET | 82.6 (0.8) → 59.8 (1.3) | 79.7 (1.6) → 56.7 (1.8) |
| vit_base_patch16_clip_224.datacompxl | OPENAI | 83.4 (1.1) → 53.6 (2.3) | 81.4 (1.0) → 50.7 (2.6) |
| vit_base_patch16_clip_224.datacompxl | INCEPTION | 84.7 (0.7) → 59.7 (4.5) | 82.8 (0.8) → 57.3 (3.8) |
| vit_base_patch16_clip_224.datacompxl | IMAGENET | 83.6 (0.9) → 52.2 (2.8) | 81.5 (1.1) → 49.3 (2.6) |
| vit_base_patch16_clip_224.dfn2b | OPENAI | 84.8 (1.2) → 58.8 (2.6) | 83.0 (1.3) → 56.4 (2.3) |
| vit_base_patch16_clip_224.dfn2b | INCEPTION | 87.3 (1.6) → 69.6 (4.7) | 86.0 (2.0) → 67.3 (5.2) |
| vit_base_patch16_clip_224.dfn2b | IMAGENET | 81.6 (2.7) → 50.0 (2.3) | 79.7 (2.8) → 48.1 (2.9) |
| vit_base_patch16_clip_224.metaclip_2pt5b | OPENAI | 83.3 (0.5) → 49.5 (3.5) | 81.1 (0.9) → 47.9 (3.2) |
| vit_base_patch16_clip_224.metaclip_2pt5b | INCEPTION | 85.8 (0.9) → 62.1 (2.0) | 83.4 (0.6) → 60.1 (1.8) |
| vit_base_patch16_clip_224.metaclip_2pt5b | IMAGENET | 81.3 (2.5) → 45.3 (4.5) | 78.7 (2.7) → 43.6 (4.2) |
| vit_base_patch16_clip_224.openai | OPENAI | 83.4 (0.5) → 60.1 (2.4) | 81.8 (0.8) → 57.7 (2.8) |
| vit_base_patch16_clip_224.openai | INCEPTION | 85.5 (0.8) → 66.7 (3.4) | 83.3 (1.3) → 65.1 (3.7) |
| vit_base_patch16_clip_224.openai | IMAGENET | 75.3 (14.1) → 50.1 (13.8) | 72.7 (13.9) → 47.5 (12.6) |
| vit_base_patch16_clip_224.laion2b | OPENAI | 81.4 (1.4) → 52.1 (2.2) | 78.5 (2.5) → 50.0 (2.0) |
| vit_base_patch16_clip_224.laion2b | INCEPTION | 84.6 (0.6) → 62.0 (2.1) | 82.2 (0.4) → 59.9 (2.1) |
| vit_base_patch16_clip_224.laion2b | IMAGENET | 81.0 (0.4) → 50.1 (0.7) | 78.7 (0.5) → 48.3 (1.1) |
| vit_base_patch16_224.augreg_in1k | OPENAI | 83.4 (0.4) → 67.6 (0.7) | 81.7 (0.8) → 65.8 (0.7) |
| vit_base_patch16_224.augreg_in1k | INCEPTION | 83.9 (0.5) → 69.3 (0.7) | 81.8 (0.3) → 67.8 (0.5) |
| vit_base_patch16_224.augreg_in1k | IMAGENET | 83.7 (0.4) → 67.5 (1.2) | 81.8 (0.2) → 65.9 (0.7) |
| vit_base_patch16_224.augreg_in21k | OPENAI | 89.6 (0.2) → 84.0 (0.5) | 88.9 (0.5) → 83.4 (0.4) |
| vit_base_patch16_224.augreg_in21k | INCEPTION | 89.6 (0.2) → 84.9 (0.7) | 88.7 (0.4) → 83.7 (0.6) |
| vit_base_patch16_224.augreg_in21k | IMAGENET | 89.5 (0.2) → 83.9 (0.1) | 88.9 (0.3) → 83.4 (0.7) |
| vit_base_patch16_224.mae | OPENAI | 76.7 (0.5) → 39.3 (4.4) | 74.1 (0.6) → 36.7 (3.9) |
| vit_base_patch16_224.mae | INCEPTION | 74.0 (0.3) → 41.1 (4.2) | 72.5 (1.1) → 38.9 (4.4) |
| vit_base_patch16_224.mae | IMAGENET | 76.4 (0.9) → 38.0 (1.5) | 74.4 (0.5) → 35.6 (1.7) |

so the per-output-pixel variance inherits the $k^{-2}$ and $s^{-2}$ scalings up to the local factor $\bar{x}$. Applying the Anscombe transform $A(y) = 2\sqrt{y + 3/8}$ approximately stabilizes the Poisson variance to $\approx 1$, after which Gaussian-based methods are applicable (Anscombe, 1948).

**Salt-and-pepper noise.** Under the symmetric model where each pixel is replaced by either $0$ or $1$ with probability $q$ and a locally constant patch with mean $\bar{x}$, we have

$$\mathbb{E}[\text{avg error}] = q(1/2 - \bar{x}), \quad \text{Var}[\text{avg error}] = O(1/m).$$

Max pooling tends to amplify these impulses. As a robust alternative, median pooling recovers the clean value in constant patches when contamination is lower than $50\%$ and is 1-Lipschitz with respect to $\ell_\infty$; trimmed means are another option.

**Normalization and Lipschitz sensitivity.** The pixel-space Lipschitz bound in Theorem 5 does not depend on the specific noise type, so smaller per-channel normalization stds increase the worst-case sensitivity equally for Gaussian and non-Gaussian perturbations.

# E  ARE THERE OTHER FACTORS THAT CAUSE VULNERABILITIES OF CLIP?

We investigated other factors that might possibly address the vulnerability of CLIP. However, the vulnerability of CLIP could not be fully addressed by other factors examined below.

**How about swapping pretrained weights with supervised ViT?** Answer: No. Differences in training datasets and losses would lead to different pretrained weights for CLIP ViTs. Assuming that certain dataset or loss properties, or equivalently certain properties of the pretrained weights of CLIP ViTs, lead to vulnerabilities, we performed controlled experiments to swap parts of them with those of supervised ViTs.

Table 13: Classification accuracy (%) for fine-tuning ViTs on the Stanford-Cars.

| Pretrained Model | Mean-Std | Val. Acc. w/ Noise | Test Acc. w/ Noise |
|---|---|---|---|
| vit_base_patch16_clip_224.openai_ft_in12k_in1k | OPENAI | 83.8 (0.1) → 71.0 (1.4) | 83.0 (0.7) → 69.7 (0.7) |
| vit_base_patch16_clip_224.openai_ft_in12k_in1k | INCEPTION | 87.3 (1.2) → 77.7 (2.1) | 86.2 (1.3) → 76.3 (2.2) |
| vit_base_patch16_clip_224.openai_ft_in12k_in1k | IMAGENET | 81.1 (1.6) → 63.8 (2.2) | 80.7 (1.9) → 64.6 (2.2) |
| vit_base_patch16_clip_224.datacompxl | OPENAI | 90.1 (0.7) → 76.1 (1.7) | 89.2 (0.6) → 75.3 (1.5) |
| vit_base_patch16_clip_224.datacompxl | INCEPTION | 91.3 (0.2) → 80.9 (0.8) | 90.4 (0.6) → 79.4 (1.0) |
| vit_base_patch16_clip_224.datacompxl | IMAGENET | 89.8 (1.4) → 75.4 (3.8) | 89.1 (1.4) → 74.3 (3.8) |
| vit_base_patch16_clip_224.dfn2b | OPENAI | 91.1 (0.5) → 78.9 (2.5) | 90.2 (0.5) → 77.8 (2.2) |
| vit_base_patch16_clip_224.dfn2b | INCEPTION | 94.2 (1.1) → 88.7 (2.2) | 93.2 (1.0) → 87.6 (2.8) |
| vit_base_patch16_clip_224.dfn2b | IMAGENET | 91.1 (1.8) → 78.8 (5.0) | 90.7 (1.4) → 77.6 (5.4) |
| vit_base_patch16_clip_224.metaclip_2pt5b | OPENAI | 87.7 (0.7) → 67.7 (1.7) | 86.9 (0.7) → 66.4 (1.7) |
| vit_base_patch16_clip_224.metaclip_2pt5b | INCEPTION | 91.1 (0.3) → 78.5 (1.3) | 90.2 (0.4) → 77.3 (1.6) |
| vit_base_patch16_clip_224.metaclip_2pt5b | IMAGENET | 87.1 (1.3) → 64.7 (2.0) | 86.1 (1.7) → 63.2 (2.3) |
| vit_base_patch16_clip_224.openai | OPENAI | 85.6 (3.3) → 73.5 (4.1) | 85.3 (3.1) → 72.4 (3.9) |
| vit_base_patch16_clip_224.openai | INCEPTION | 89.8 (0.4) → 81.0 (1.1) | 89.5 (0.4) → 80.2 (0.7) |
| vit_base_patch16_clip_224.openai | IMAGENET | 85.2 (1.6) → 70.1 (3.0) | 84.2 (1.3) → 69.0 (3.0) |
| vit_base_patch16_clip_224.laion2b | OPENAI | 84.8 (2.4) → 65.6 (4.2) | 84.1 (2.3) → 65.3 (3.7) |
| vit_base_patch16_clip_224.laion2b | INCEPTION | 89.9 (0.8) → 78.4 (2.3) | 88.8 (0.9) → 77.0 (2.2) |
| vit_base_patch16_clip_224.laion2b | IMAGENET | 79.9 (4.7) → 54.5 (6.6) | 79.5 (5.1) → 54.9 (7.6) |
| vit_base_patch16_224.augreg_in1k | OPENAI | 82.8 (0.5) → 67.4 (1.0) | 81.6 (0.4) → 66.3 (0.9) |
| vit_base_patch16_224.augreg_in1k | INCEPTION | 83.2 (0.6) → 69.2 (1.1) | 81.6 (0.5) → 67.5 (1.3) |
| vit_base_patch16_224.augreg_in1k | IMAGENET | 83.0 (0.3) → 66.2 (1.4) | 81.5 (0.2) → 65.1 (1.6) |
| vit_base_patch16_224.augreg_in21k | OPENAI | 89.7 (0.2) → 82.6 (0.5) | 88.5 (0.3) → 81.4 (0.5) |
| vit_base_patch16_224.augreg_in21k | INCEPTION | 89.9 (0.2) → 84.2 (0.4) | 88.3 (0.3) → 83.3 (0.7) |
| vit_base_patch16_224.augreg_in21k | IMAGENET | 89.9 (0.5) → 81.9 (0.5) | 88.6 (0.6) → 81.1 (0.3) |
| vit_base_patch16_224.mae | OPENAI | 80.4 (0.5) → 61.1 (1.5) | 78.0 (0.6) → 58.5 (0.9) |
| vit_base_patch16_224.mae | INCEPTION | 80.3 (0.3) → 61.7 (1.0) | 77.6 (0.5) → 59.3 (0.8) |
| vit_base_patch16_224.mae | IMAGENET | 80.6 (0.4) → 58.1 (2.2) | 78.3 (0.3) → 56.7 (2.4) |

Specifically, we swapped pretrained weights of each block in vit_base_patch16_clip_224.openai with those of vit_base_patch16_224.augreg2_in21k_ft_in1k to see which module weights determine the robustness against Gaussian noise (Table 14). Although swapping pretrained weights partially addressed the vulnerability of CLIP ViTs in certain cases near the last block such as targeting block12, the improvements were not as significant as the approach of replacing mean-std constants. Furthermore, the improvement depended on the specific weight choice in the target block; block12.mlp.fc2.weight improved robustness, whereas block12.norm1.weight did not. When we swapped multiple weights such as block12.{mlp.fc2, mlp.fc1, norm2}, the performance rather degraded, which indicates that improvement is not guaranteed.

**How about architectural differences such as norm_pre?** Answer: No. Although the architecture is almost the same for CLIP ViT and supervised ViTs, one difference is that CLIP ViTs insert additional LayerNorm in the patch embedding before the transformer blocks start, which we refer to as norm_pre. Assuming that the use of norm_pre causes vulnerability, we performed controlled experiments training ViTs with and without norm_pre (Table 16). Nevertheless, the ViT with norm_pre, which corresponds to the identical architecture of CLIP ViTs, rather exhibited improved performance against Gaussian noise, which indicates that norm_pre does not lead to the vulnerability observed in CLIP ViTs.

## F EMPIRICAL SIMULATIONS FOR TESTING ASSUMPTION AND THEOREMS

We performed module-level simulations to compare empirical results with the expected values stated in the assumption and theorems. All simulation results closely matched the theoretical expectations. The used Python source code is available in the supplementary materials.

Table 14: Results of swapping pretrained weights in CLIP ViT. The accuracy with Gaussian noise partially improved.

| Swap | Val. Acc. $\rightarrow$ w/ Noise | Test Acc. $\rightarrow$ w/ Noise |
|---|---|---|
| stem | 53.1 (1.2) $\rightarrow$ 36.3 (0.9) | 51.3 (1.5) $\rightarrow$ 35.4 (1.2) |
| block1 | 81.3 (19.1) $\rightarrow$ 48.6 (14.8) | 80.1 (19.9) $\rightarrow$ 46.2 (14.1) |
| block2 | 41.9 (3.4) $\rightarrow$ 20.1 (1.6) | 40.7 (3.9) $\rightarrow$ 18.8 (1.6) |
| block3 | 68.6 (13.6) $\rightarrow$ 29.9 (4.5) | 67.9 (13.7) $\rightarrow$ 28.7 (4.4) |
| block4 | 80.2 (6.8) $\rightarrow$ 29.6 (7.5) | 79.8 (6.4) $\rightarrow$ 28.6 (7.9) |
| block5 | 77.0 (4.0) $\rightarrow$ 30.0 (4.0) | 77.3 (3.2) $\rightarrow$ 29.0 (3.4) |
| block6 | 84.7 (0.8) $\rightarrow$ 39.9 (1.9) | 84.0 (0.5) $\rightarrow$ 37.8 (2.2) |
| block7 | 87.8 (0.4) $\rightarrow$ 45.0 (0.8) | 86.5 (0.6) $\rightarrow$ 44.9 (1.2) |
| block8 | 90.5 (0.4) $\rightarrow$ 49.9 (2.7) | 88.4 (0.6) $\rightarrow$ 48.4 (1.6) |
| block9 | 90.7 (0.2) $\rightarrow$ 56.5 (3.9) | 90.1 (0.4) $\rightarrow$ 54.8 (2.2) |
| block10 | 91.5 (0.4) $\rightarrow$ 62.3 (3.3) | 91.0 (0.4) $\rightarrow$ 60.4 (3.5) |
| block11 | 91.4 (0.4) $\rightarrow$ 59.6 (5.1) | 90.7 (0.9) $\rightarrow$ 58.1 (6.0) |
| block12 | 91.5 (0.5) $\rightarrow$ 62.3 (4.9) | 91.4 (0.6) $\rightarrow$ 60.8 (4.4) |
| head | 82.8 (7.6) $\rightarrow$ 48.4 (7.5) | 82.3 (6.8) $\rightarrow$ 48.2 (6.6) |
| Baseline (IMAGENET) | 91.2 (0.5) $\rightarrow$ 58.5 (4.0) | 90.7 (0.8) $\rightarrow$ 58.4 (4.3) |
| Ours (INCEPTION) | 92.5 (0.3) $\rightarrow$ 71.7 (1.0) | 91.9 (0.6) $\rightarrow$ 70.2 (1.2) |

Table 15: Results of swapping specific weights in block12. Swapping multiple weights did not ensure improved robustness.

| Swap | Val. Acc. $\rightarrow$ w/ Noise | Test Acc. $\rightarrow$ w/ Noise |
|---|---|---|
| block12.norm1.weight | 91.0 (1.3) $\rightarrow$ 56.8 (8.2) | 90.1 (1.3) $\rightarrow$ 56.6 (8.6) |
| block12.norm1.bias | 91.5 (0.9) $\rightarrow$ 59.5 (5.7) | 90.7 (1.4) $\rightarrow$ 58.1 (6.1) |
| block12.attn.qkv.weight | 91.0 (0.5) $\rightarrow$ 60.5 (1.3) | 90.3 (0.8) $\rightarrow$ 59.0 (1.4) |
| block12.attn.qkv.bias | 91.0 (0.9) $\rightarrow$ 58.7 (3.6) | 90.0 (0.9) $\rightarrow$ 58.4 (3.7) |
| block12.attn.proj.weight | 92.1 (0.6) $\rightarrow$ 59.8 (5.6) | 91.3 (0.9) $\rightarrow$ 59.8 (5.8) |
| block12.attn.proj.bias | 90.9 (1.0) $\rightarrow$ 58.3 (5.4) | 90.2 (1.2) $\rightarrow$ 57.8 (5.0) |
| block12.norm2.weight | 91.8 (0.7) $\rightarrow$ 62.7 (4.3) | 90.7 (0.9) $\rightarrow$ 61.1 (3.7) |
| block12.norm2.bias | 91.4 (0.9) $\rightarrow$ 60.8 (3.4) | 90.8 (0.7) $\rightarrow$ 59.7 (3.4) |
| block12.mlp.fc1.weight | 91.4 (1.0) $\rightarrow$ 61.0 (7.6) | 91.0 (1.4) $\rightarrow$ 60.3 (7.0) |
| block12.mlp.fc1.bias | 91.2 (1.3) $\rightarrow$ 58.3 (3.4) | 90.4 (1.5) $\rightarrow$ 57.4 (3.7) |
| block12.mlp.fc2.weight | 91.3 (0.4) $\rightarrow$ 65.2 (2.2) | 90.5 (0.3) $\rightarrow$ 63.8 (2.1) |
| block12.mlp.fc2.bias | 91.4 (0.7) $\rightarrow$ 58.8 (5.0) | 90.7 (0.7) $\rightarrow$ 58.2 (5.4) |
| block12.mlp.fc2 | 90.8 (0.6) $\rightarrow$ 58.7 (3.2) | 90.0 (0.4) $\rightarrow$ 57.7 (3.1) |
| block12.mlp.fc2 & mlp.fc1 | 91.9 (1.1) $\rightarrow$ 64.2 (4.3) | 91.6 (0.8) $\rightarrow$ 63.8 (5.0) |
| block12.mlp.fc2 & mlp.fc1 & norm2 | 91.0 (0.5) $\rightarrow$ 55.5 (4.3) | 90.0 (1.4) $\rightarrow$ 54.0 (5.2) |

$\mathbf{A_{roll}}$ We embed each $k \times k$ kernel into a $512 \times 512$ grid, compute the normalized spectrum $|\widehat{K}|$, form its $\ell_2$-radial profile, and fit the low-pass envelope $\phi_k(r) = (1 + \beta k r)^{-(1+\delta)}$ by weighted log-MSE (Table 17). For representative radii of $\pi/8, \pi/4, \pi/2$, we observed that the empirical magnitudes lie below the fitted envelopes, which verifies this assumption in practice.

Table 16: Results on different ViT architectures with and without norm_pre. The use of norm_pre did not bring vulnerability.

| Architecture | Top-1 → w/ Noise | Top-5 → w/ Noise |
|---|---|---|
| w/o norm_pre | 77.76 → 47.15 | 93.84 → 68.96 |
| w/ norm_pre | 78.84 → 54.22 | 94.14 → 76.13 |

Table 17: The upper block reports the results for the box kernel. The lower block reports the results for the Gaussian kernel.

| $r$ (rad) | Empirical ($|\widehat{K}_k(\boldsymbol{\omega})|$) | Theoretical ($\phi_k(\|\boldsymbol{\omega}\|)$) |
|---|---|---|
| 0.3962 | 0.0297134 | 0.0570019 |
| 0.7886 | 0.0129235 | 0.0167515 |
| 1.5661 | 0.0040941 | 0.0042482 |
| 0.3962 | 0.0226295 | 0.0326660 |
| 0.7886 | 0.0059380 | 0.0068950 |
| 1.6031 | 0.0007060 | 0.0010978 |

Table 18: Measured $\gamma$ for a $k \times k$ kernel. Stds for 100 simulations are reported.

| $k$ | Empirical | Theoretical |
|---|---|---|
| 4 | $0.062535 \pm 0.001019$ | 0.062500 |
| 8 | $0.015584 \pm 0.000484$ | 0.015625 |
| 12 | $0.006911 \pm 0.000296$ | 0.006944 |
| 16 | $0.003888 \pm 0.000212$ | 0.003906 |
| 20 | $0.002487 \pm 0.000169$ | 0.002500 |
| 24 | $0.001728 \pm 0.000148$ | 0.001736 |
| 28 | $0.001271 \pm 0.000129$ | 0.001276 |
| 32 | $0.000973 \pm 0.000115$ | 0.000977 |

**Theorem 2** Table 18 reports the Monte Carlo estimate of the per-pixel noise gain $\gamma$ for a $k \times k$ normalized box filter. We convolve i.i.d. $\mathcal{N}(0, \sigma^2)$ noise with the filter via FFT-based circular convolution and compare the empirical $\hat{\gamma}$ with the theoretical $\|K_k\|_F^2 = 1/k^2$, where $K_k$ is the normalized $k \times k$ box stem kernel.

**Theorem 3** Table 19 reports Monte Carlo estimates of the per-output-pixel noise gain $\gamma_\downarrow(s)$ under anti-aliased downsampling by a factor $s$, using a $g(s) \times g(s)$ normalized box prefilter and decimation. We compare the empirical $\hat{\gamma}_\downarrow$ with the theoretical $\|K_{g(s)}\|_F^2 = 1/g(s)^2$, implying $\sim s^{-2}$ when $g(s) \propto s$.

**Theorem 4** The results in Table 20 were obtained via Monte Carlo with 200k trials on $S + \eta$ with $\eta \sim \mathcal{N}(0, 1)$ and $k = w^2$. Theoretical entries correspond to $\sigma^2/k$ for average pooling and Gauss-Hermite quadrature for $E[M_k]$ and $E[M_k^2]$ to compute max-pooling bias and MSE.

**Theorem 5** We construct random linear maps $A$ with $\|A\|_2 = L_z = 3.0$, compose them with $D = \text{diag}(1/\boldsymbol{\sigma})$ from INCEPTION and OPENAI, and estimate $\|AD\|_2$ via power iteration. Table 21 compares the theoretical bound $L_z/\sigma_{\min}$ with the measured norm and their ratio, confirming the predicted $1/\sigma_{\min}$ scaling.

Table 19: Measured $\gamma_\downarrow(s)$ for anti-aliased downsampling by a factor of $s$. Stds for 100 simulations are reported.

| $s$ | Empirical | Theoretical |
|---|---|---|
| 1 | $0.999667 \pm 0.006170$ | 1.000000 |
| 2 | $0.250114 \pm 0.002767$ | 0.250000 |
| 3 | $0.110970 \pm 0.001869$ | 0.111111 |
| 4 | $0.062447 \pm 0.001483$ | 0.062500 |
| 6 | $0.027772 \pm 0.000880$ | 0.027778 |
| 8 | $0.015567 \pm 0.000649$ | 0.015625 |
| 12 | $0.006945 \pm 0.000433$ | 0.006944 |
| 16 | $0.003925 \pm 0.000359$ | 0.003906 |

Table 20: Comparison of empirical (Em.) and theoretical (Th.) results for average and max poolings

| $w$ | $k$ | Avg MSE (Em.) | Avg MSE (Th.) | Max Bias (Em.) | Max Bias (Th.) | Max MSE (Em.) | Max MSE (Th.) |
|---|---|---|---|---|---|---|---|
| 2 | 4 | 0.25083 | 0.25000 | 1.02936 | 1.02938 | 1.55372 | 1.55133 |
| 3 | 9 | 0.11049 | 0.11111 | 1.48535 | 1.48501 | 2.56409 | 2.56262 |
| 4 | 16 | 0.06265 | 0.06250 | 1.76524 | 1.76599 | 3.41148 | 3.41374 |
| 5 | 25 | 0.04006 | 0.04000 | 1.96619 | 1.96531 | 4.12369 | 4.12097 |
| 6 | 36 | 0.02779 | 0.02778 | 2.11722 | 2.11812 | 4.71818 | 4.72069 |

Table 21: Measured $\|AD\|_2$ closely matches the bound $L_z/\sigma_{\min}$ for random $A$ under INCEPTION and OPENAI, confirming the $1/\sigma_{\min}$ scaling

| Constants | Bound $L_z/\sigma_{\min}$ | Measured $\|AD\|_2$ | $\dfrac{L_z/\sigma_{\min}}{\|AD\|_2}$ |
|---|---|---|---|
| INCEPTION | 6.000000 | 5.998213 | 1.000298 |
| OPENAI | 11.480943 | 11.200055 | 1.025079 |

## G  RANK DIFFERENCE AS A ROBUSTNESS PROXY

Here, we denote the rank difference (RankDiff) at severity $\tau > 0$,

$$\text{RankDiff}_i(\tau) \coloneqq \text{rank}_\tau(i) - \text{rank}_0(i),$$

where $\text{rank}_\tau$ orders models by accuracy at $\tau$, so a more negative $\text{RankDiff}_i$ indicates a robustness gain. In this section, we show that RankDiff is a principled, scale-free proxy because it aggregates pairwise rank flips caused by robustness slope differences.

**Assumption (local linearity with quadratic remainder).**  For model $i \in \{1, \dots, M\}$, let $A_i(\tau)$ be its accuracy at noise severity $\tau \geq 0$ and $p_i \coloneqq A_i(0)$. For some $\tau_0 > 0$,

$$A_i(\tau) = p_i - \rho_i\tau + r_i(\tau), \qquad \rho_i \geq 0, \qquad |r_i(\tau)| \leq L_i\tau^2 \quad (\tau \in [0, \tau_0]), \tag{11}$$

where $\rho_i$ is the first-order robustness slope, and $L_i$ bounds the curvature. The linear accuracy drop after applying a specific corruption has been verified in several studies (Recht et al., 2019; Hendrycks & Dieterich, 2019).

**Pairwise flip rule.**  For any $i \neq j$,

$$A_i(\tau) - A_j(\tau) = (p_i - p_j) - (\rho_i - \rho_j)\tau + \varepsilon_{ij}(\tau), \qquad |\varepsilon_{ij}(\tau)| \leq (L_i + L_j)\tau^2. \tag{12}$$

If $\rho_i \neq \rho_j$, the first-order flip threshold is

$$\tau_{ij}^\star := \frac{p_i - p_j}{\rho_i - \rho_j}. \tag{13}$$

When $\tau_{ij}^\star \in (0, \tau_0]$ and the margin condition

$$|(p_i - p_j) - (\rho_i - \rho_j)\tau| > (L_i + L_j)\tau^2 \tag{14}$$

holds at $\tau$, the sign of $A_i(\tau) - A_j(\tau)$ is determined by the first-order term: Model $i$ outranks $j$ at $\tau$ if and only if $\tau > \tau_{ij}^\star$ when $\rho_i < \rho_j$ (Figure 5).

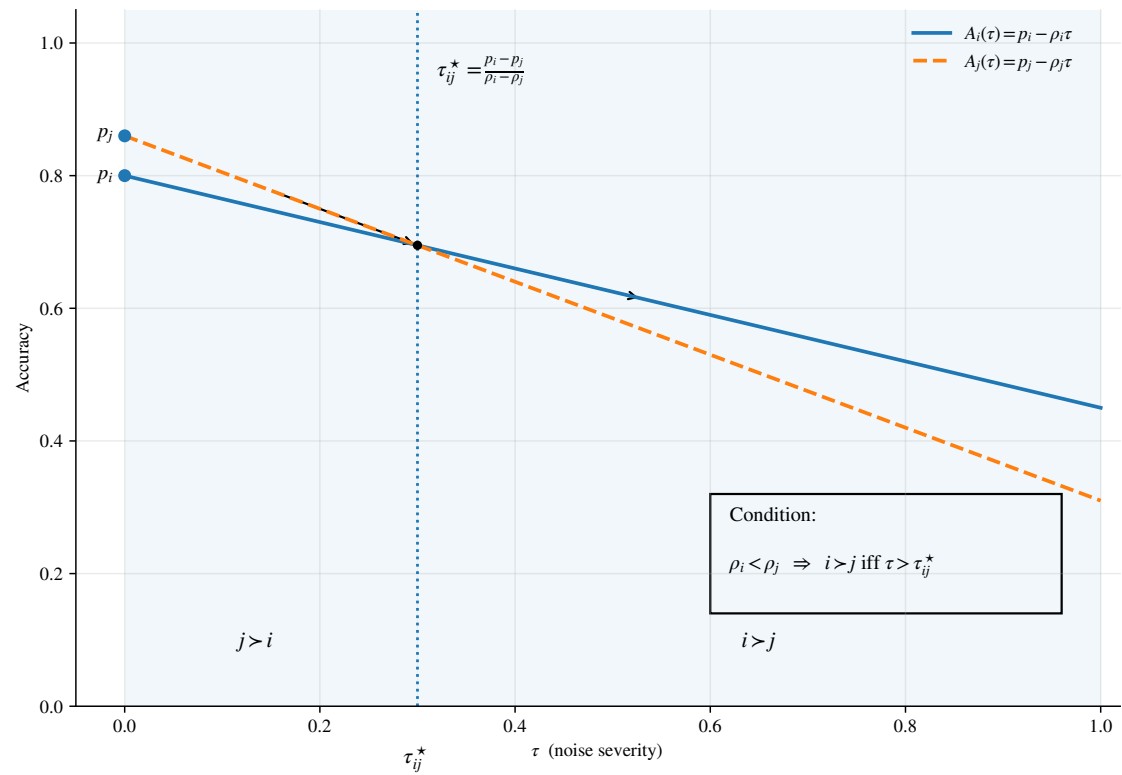

Figure 5: Illustration of a rank flip

**RankDiff counts robustness-driven flips.** Let $\mathcal{A}_i(\tau) := \{j \neq i : |(p_i - p_j) - (\rho_i - \rho_j)\tau| \leq (L_i + L_j)\tau^2\}$ be the set of ambiguous pairs at $\tau$. Then, we have

$$\left| \text{RankDiff}_i(\tau) + \sum_{j \neq i} \text{sgn}(\rho_j - \rho_i)\mathbf{1}\{\rho_i \neq \rho_j, 0 < \tau_{ij}^\star \leq \tau\} \right| \leq |\mathcal{A}_i(\tau)|. \tag{15}$$

In particular, if Eq. 14 holds for all $j \neq i$ at $\tau$, equality holds in Eq. 15: $\text{RankDiff}_i(\tau)$ equals the net number of pairwise flips caused by having a smaller slope $\rho_i$.

Under the empirically observed near-linearity of accuracy-severity curves within the tested range, RankDiff is a scale-free robustness score: It ignores absolute calibration of accuracies and rewards models with smaller slopes $\rho_i$ by counting the robustness-driven improvements in relative order.

Although we wrote that a negative RankDiff indicates better robustness, we are not saying that understanding the absolute value of RankDiff would capture robustness; to clarify our approach, we rather compare pairwise architectures to compute their corresponding $\Delta$RankDiff to understand the relative difference in robustness.

## H   EXPERIMENTAL SETUP

Here, we present the experimental details and full hyperparameters for the implementations.

**Gaussian Noise**   We injected Gaussian noise into images using the `GaussNoise()` function from the `Albumentations` library (Buslaev et al., 2020). By default, we used the transform `A.GaussNoise(std_range=(0.1, 0.22), p=1.0)` with a scale factor with range (0.1, 0.22), which determines the fraction of the maximum value, *i.e.*, 255 for uint8 images or 1.0 for float images. For ImageNet-1K experiments, we used a scale factor with a range of (0.2, 0.44). The probability of applying Gaussian noise was set to 1. Note that Gaussian noise was applied only during evaluation, *i.e.*, during the test phase, not during the training phase.

**ResNet Experiments**   We targeted multi-class classification tasks on the Oxford-IIIT Pet, Caltech-101, FGVC-Aircraft, Caltech-UCSD Birds-200-2011, and Stanford Cars datasets. The Oxford-IIIT Pet dataset contains 7K pet images from 37 classes; the Caltech-101 dataset includes 9K object images from 101 classes with a background category; the FGVC-Aircraft dataset includes 10K aircraft images from 102 classes; the Caltech-UCSD Birds-200-2011 dataset includes 12K bird images from 200 classes; and the Stanford Cars dataset includes 16K car images from 196 classes. These datasets are publicly available on their official websites. Each dataset was split into training, validation, and test sets with a ratio of 70:15:15. Unless specified otherwise, all experiments were conducted at a resolution of $224^2$ using standard data augmentation, including random resized cropping to 256 pixels, random rotations within 15 degrees, color jitter with a factor of 0.4, random horizontal flip with a probability of 0.5, center cropping with 224-pixel windows, and mean-std normalization based on ImageNet statistics.

For training, stochastic gradient descent with a momentum of 0.9, learning rate of 0.01, cosine annealing schedule with 200 iterations (Loshchilov & Hutter, 2017), weight decay of $10^{-2}$, and mini-batch size of 128 were used. These hyperparameters were determined based on the accuracy of the validation set. One exception was made for experiments with larger resolutions ranging from $224^2$ to $896^2$, where we used mini-batch size of 64 to adjust GPU memory, while other hyperparameters are the same. The model with the highest validation accuracy was obtained after 200 training epochs, and we reported accuracy on the validation and test sets. The ResNets were trained from scratch to solely focus on the architectural difference. The training was conducted on a single GPU machine. An average and standard deviation of five runs with different random seeds were reported for each result.

For ResNet, we used five types with the following architectures:

- Original ResNet: $7 \times 7$ stem with a width = 64 with single-layer, strided convolution in downsampling.
- ResNet-C: 3-layer $3 \times 3$ stem with a width = 32 (32, 32, 64), strided convolution in downsampling.
- ResNet-D: 3-layer $3 \times 3$ stem with a width = 32 (32, 32, 64), average pool in downsampling.
- ResNet-S: 3-layer $3 \times 3$ stem with a width = 64 (64, 64, 128), strided convolution in downsampling.

- ResNet-T: 3-layer $3 \times 3$ stem with a width = 32 (24, 48, 64), average pool in downsampling.

**CLIP Experiments** For the CLIP experiments, we used pretrained weights for both supervised ViTs and CLIP ViTs. When performing fine-tuning experiments, we used a learning rate of 0.001 and a weight decay of $2 \times 10^{-4}$, while keeping all other hyperparameters the same as in the above setup in ResNet.

**ImageNet-1K Training** The ImageNet-1K dataset contains 1.28M images for 1,000 classes. We referred to the hyperparameter recipe described in the official documentation and the recipe from DeiT (Touvron et al., 2021). For training, the AdamW optimizer (Loshchilov & Hutter, 2019) with learning rate $5 \times 10^{-4}$, epochs 400, warm-up learning rate $10^{-6}$, cosine annealing schedule (Loshchilov & Hutter, 2017), weight decay 0.05, label smoothing (Szegedy et al., 2016) 0.1, RandAugment (Cubuk et al., 2020) of magnitude 9 and noise-std 0.5 with increased severity (rand-m9-mstd0.5-inc1), random erasing (Zhong et al., 2020) with probability 0.25, Cutmix (Yun et al., 2019) 1.0, stochastic depth (Huang et al., 2016) 0.1, mini-batch size 128 per GPU, Exponential Moving Average of model weights with decay factor 0.99996, and image resolution $224^2$ were used. The training was performed on a $4 \times$A100 GPU machine, which required two to three days per training.

**Mean-Std Constants** Note that pretrained models may have been trained by any of the normalization constants; our choice of mean-std constants was applied on evaluation or fine-tuning of pretrained models. For training our own models, mean-std constants were applied during both the training and test phases. The exact values are as follows:

```
OPENAI_CLIP_MEAN = (0.48145466, 0.4578275, 0.40821073)
OPENAI_CLIP_STD = (0.26862954, 0.26130258, 0.27577711)
IMAGENET_INCEPTION_MEAN = (0.5, 0.5, 0.5)
IMAGENET_INCEPTION_STD = (0.5, 0.5, 0.5)
IMAGENET_DEFAULT_MEAN = (0.485, 0.456, 0.406)
IMAGENET_DEFAULT_STD = (0.229, 0.224, 0.225)
```

# I LIST OF NOTATIONS

Table 22: Kernel and resolution-related notations.

| Symbol | Description |
|---|---|
| $x \in [0,1]^{C \times H \times W}$ | Input image with $C$ channels, height $H$, width $W$. |
| $\eta \sim \mathcal{N}(0, \sigma^2 I)$ | Additive i.i.d. Gaussian noise with per-pixel std $\sigma$. |
| $I, I_n$ | Identity matrix of appropriate size; $I_n \in \mathbb{R}^{n \times n}$. |
| $*$ | 2D discrete convolution. |
| $\widehat{u}$ | DFT of $u$ on the grid $\Omega$. |
| $\Omega$ | DFT grid. |
| $\varepsilon$ | Infrared cutoff $\varepsilon = 2\pi / \max\{H, W\}$. |
| $K_k \in \mathbb{R}^{k \times k}$ | Stem kernel of side length $k$; $\widehat{K}_k$ denotes its DFT. |
| $\phi_k(r) = (1 + \beta k r)^{-1-\delta}$ | Radial low-pass envelope upper-bounding $|\widehat{K}_k(\omega)|$. |
| $\beta, \delta$ | Positive envelope constants. |
| $\gamma(k) = \dfrac{\mathbb{E}\|K_k * \eta\|_2^2}{\sigma^2 HW}$ | Per-pixel noise gain of the stem; equals $\|K_k\|_F^2$. |
| $s \geq 1$ | Downsampling factor. |
| $g(s)$ | Anti-alias filter size before downsampling; $c_1 s \leq g(s) \leq c_2 s$. |
| $c_1, c_2$ | Absolute positive constants, independent of $s$. |
| $D_s = (\Downarrow_s) \circ K_{g(s)}$ | Anti-aliased downsampling: Filter then downsample by $s$. |
| $\Downarrow_s$ | Downsampling by a factor $s$ along height and width. |
| $\gamma_\downarrow(s) = \dfrac{\mathbb{E}\|D_s \eta\|_2^2}{\sigma^2 HW/s^2}$ | Per-output-pixel noise gain after downsampling. |
| $\mathbb{E}[\cdot], \mathrm{Var}[\cdot]$ | Expectation and variance. |
| $C, C'$ | Absolute constants independent of $k$ and $s$ in the bounds. |
| $\|\cdot\|_2, \|\cdot\|_\infty, \|\cdot\|_F$ | Euclidean, sup, and Frobenius norms. |

Table 23: Pooling and CLIP-related notations.

| Symbol | Description |
|---|---|
| $w, m = w^2$ | Pooling window side length and number of elements. |
| $S = (S_1, \ldots, S_m)$ | Clean activations in one pooling window; $S_{(j)}$ denotes the $j$-th order statistic. |
| $X_{\mathrm{avg}} = \frac{1}{m} \sum_{i=1}^m (S_i + \eta_i)$ | Average-pooled noisy activation. |
| $X_{\max} = \max_{1 \leq i \leq m}(S_i + \eta_i)$ | Max-pooled noisy activation. |
| $S_{\mathrm{avg}} = \frac{1}{m} \sum_i S_i, \; S_{\max} = \max_i S_i$ | Clean pooled activations. |
| $\delta_{\mathrm{avg}} = X_{\mathrm{avg}} - S_{\mathrm{avg}}$ | Avg-pool error; $\mathbb{E}[\delta_{\mathrm{avg}}] = 0$, $\mathrm{Var}[\delta_{\mathrm{avg}}] = \sigma^2/m$. |
| $\delta_{\max} = X_{\max} - S_{\max}$ | Max-pool error. |
| $T_{\mathrm{avg}}, T_{\max}$ | Pooling maps on a window for average and max. |
| $\|T\|_{\ell_2 \to \ell_2}$ | Lipschitz constant in $\ell_2$; $\|T_{\mathrm{avg}}\| = m^{-1/2}$, $\|T_{\max}\| \leq 1$. |
| $\Delta = S_{(1)} - S_{(2)}$ | Gap between the largest and second-largest clean entries. |
| $Z_i \overset{\text{i.i.d.}}{\sim} \mathcal{N}(0,1)$ | Standard normals; $M_m = \max_i Z_i$, $A_m = \max_i |Z_i|$. |
| $\mu \in \mathbb{R}^C, \boldsymbol{\sigma} \in \mathbb{R}_{>0}^C$ | Per-channel mean and std for input normalization. |
| $N_{\mu, \boldsymbol{\sigma}}(x) = (x - \mu)/\boldsymbol{\sigma}$ | Channel-wise normalization. |
| $f$ | Vision backbone operating on normalized inputs. |
| $z = N_{\mu, \boldsymbol{\sigma}}(x)$ | Normalized input. |
| $L_z$ | Global $\ell_2$-Lipschitz constant of $f$ on its domain. |
| $F_{\mu, \boldsymbol{\sigma}} = f \circ N_{\mu, \boldsymbol{\sigma}}$ | End-to-end map; $\|F_{\mu, \boldsymbol{\sigma}}\|_{\mathrm{Lip}} \leq L_z/\sigma_{\min}$. |
| $\sigma_{\min} = \min_c \boldsymbol{\sigma}_c$ | Smallest channel std in normalization. |

Table 24: Rank difference-related notations.

| Symbol | Description |
|---|---|
| $\tau \geq 0$ | Noise severity level. |
| $A_i(\tau)$ | Accuracy of model $i$ at severity $\tau$; $p_i = A_i(0)$ denotes clean accuracy. |
| $\rho_i$ | First-order accuracy slope with respect to severity. |
| $L_i, \tau_0$ | Curvature bound and validity radius for the local model. |
| $\mathrm{rank}_\tau(i)$ | Rank of model $i$ by accuracy at severity $\tau$. |
| $\mathrm{RankDiff}_i(\tau) = \mathrm{rank}_\tau(i) - \mathrm{rank}_0(i)$ | Rank change. |
| $\tau_{ij}^\star = \dfrac{p_i - p_j}{\rho_i - \rho_j}$ | First-order crossing severity of models $i$ and $j$. |
| $\mathcal{A}_i(\tau)$ | Set of $j$ whose ordering with $i$ is ambiguous at $\tau$. |
| $\mathrm{sgn}(\cdot), \mathbf{1}\{\cdot\}$ | Sign and indicator functions. |

## J ON GAUSSIAN NOISE

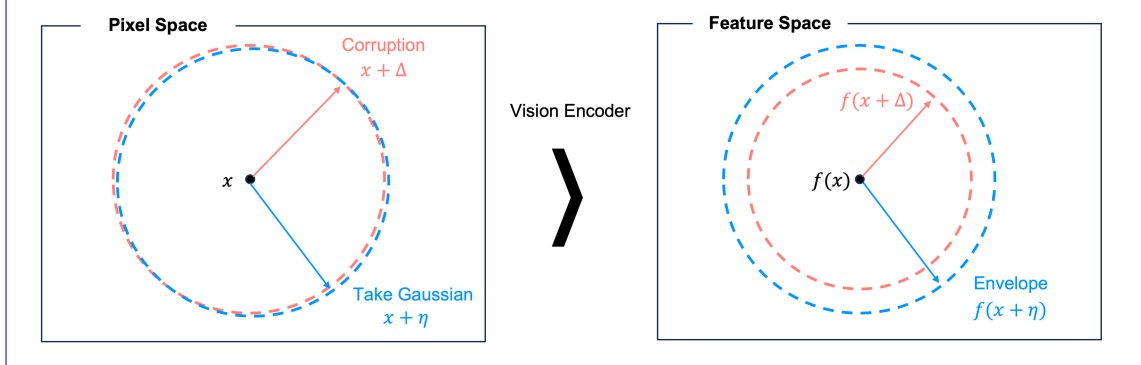

Figure 6: Illustration of Theorem 1. Features with Gaussian noise serve as an upper bound for corruption that has the same scale in pixel space.

There are several reasons why analyzing robustness against Gaussian noise is both useful and representative of common corruptions.

**Gaussian surrogate via moment matching.** The local linearization Eq. 1 implies that the feature perturbation $\delta_f$ is well-approximated to first order in the input perturbation $\Delta$. Let $\mu_\Delta := \mathbb{E}[\Delta]$ and $\Sigma_\Delta := \mathrm{Cov}(\Delta)$, where the expectation is taken over the randomness of the corruption. Plugging Eq. 1 into $\delta_f$ and taking expectations gives

$$\mathbb{E}[\delta_f] = J_f(x)\mu_\Delta + O\big(\mathbb{E}\|\Delta\|_2^2\big),$$

$$\mathrm{Cov}[\delta_f] = J_f(x)\Sigma_\Delta J_f(x)^\top + O\big(\mathbb{E}\|\Delta\|_2^3\big).$$

Thus, up to higher-order terms, any corruption whose pixel perturbation has mean $\mu_\Delta$ and covariance $\Sigma_\Delta$ induces the same first two feature-space moments as the Gaussian feature perturbation $J_f(x)\eta$ generated by $\eta \sim \mathcal{N}(\mu_\Delta, \Sigma_\Delta)$. This applies both to zero-mean corruptions, such as noise and blur, and to mean-shifting ones, such as brightness enhancement, after decomposing $\Delta$ into its mean and zero-mean residual. In this sense, Gaussian noise serves as a convenient surrogate once we restrict attention to the low-order moments of the feature perturbation.

**Gaussian probes for locally linear corruptions.** We now show that Gaussian probes are, in fact, the worst-case within a broad variance-bounded family. Let a corruption $\mathcal{C}$ with randomness $\xi$ produce $x' = \mathcal{C}(x, \xi)$ with $\Delta_{\mathcal{C}} := x' - x$, and assume that, for small severities, it admits a factorization

$$\Delta_{\mathcal{C}} = B_{\mathcal{C}}(x)\zeta, \quad \mathbb{E}[\zeta] = 0, \quad \text{Cov}(\zeta) \preceq I_d,$$

for some linear operator $B_{\mathcal{C}}(x)$ that depends smoothly on $x$ and a random vector $\zeta$. The covariance bound simply constrains the overall severity of the corruption. This model covers many image corruptions: Gaussian blur and motion blur correspond to convolutional $B_{\mathcal{C}}(x)$; brightness, contrast, and fog are locally affine rescalings; and JPEG compression or elastic distortions can be approximated as linear maps plus higher-order residuals at low severity.

Under the local linearization Eq. 1, we have $f(x + \Delta_{\mathcal{C}}) - f(x) \approx J_f(x)B_{\mathcal{C}}(x)\zeta$, and hence

$$\mathbb{E}\big[\|f(x + \Delta_{\mathcal{C}}) - f(x)\|_2^2\big] \approx \mathbb{E}\big[\|J_f(x)B_{\mathcal{C}}(x)\zeta\|_2^2\big] = \text{tr}\big(J_f(x)\Sigma_{\mathcal{C}}(x)J_f(x)^\top\big),$$

with $\Sigma_{\mathcal{C}}(x) := B_{\mathcal{C}}(x)\text{Cov}(\zeta)B_{\mathcal{C}}(x)^\top \preceq B_{\mathcal{C}}(x)B_{\mathcal{C}}(x)^\top$. Replacing $\zeta$ by $\eta \sim \mathcal{N}(0, I_d)$ yields

$$\mathbb{E}\big[\|f(x + B_{\mathcal{C}}(x)\eta) - f(x)\|_2^2\big] = \|J_f(x)B_{\mathcal{C}}(x)\|_F^2,$$

which saturates the same variance-bounded envelope: Any other zero-mean $\zeta$ with $\text{Cov}(\zeta) \preceq I_d$ can only decrease this expectation. Thus, once a corruption is reduced to a linear shape $B_{\mathcal{C}}(x)$, additive Gaussian noise with matching $B_{\mathcal{C}}(x)$ provides a worst-case, direction-agnostic stress test on $f$. Our architectural conclusions, such as kernel size, resolution, pooling, and normalization constants, depend only on how they scale this Jacobian-based quantity, so they transfer directly from Gaussian probes to a broad range of common corruptions that admit such local linear models.

**Empirical Simulation** Here, we performed an empirical simulation to investigate the validity of Theorem 1. Using common image corruptions, including blur, weather, and digital corruptions used in Hendrycks & Dietterich (2019), we first calibrated each corruption and Gaussian noise to have an equal maximum eigenvalue in pixel space and then compared the variance in feature space when passing through the same linear stem. For the stem, we considered five setups with different kernel sizes of 3, 5, and 7; high and low resolutions; and average pooling. Theoretically, Gaussian noise achieves an upper bound on this variance, and our simulations support this expectation: the ratio of corruption to Gaussian in feature-space variance saturates around 1 across all corruptions tested here (Figure 7). These results clearly demonstrate that the analysis of Gaussian noise captures the worst-case robustness against these common image corruptions.

## K    PROOF OF THEOREM 1

In this section, we prove the Gaussian envelope result stated in Theorem 1. Throughout, we fix $x$ and write $J := J_f(x)$ for the Jacobian of $f$ at $x$, and all expectations are taken with respect to the perturbation.

*Proof.* By the local linearization in Eq. 1, for any perturbation $\Delta$, we have

$$f(x + \Delta) - f(x) = J\Delta + r(x, \Delta), \qquad \|r(x, \Delta)\|_2 \leq \tfrac{L(x)}{2}\|\Delta\|_2^2, \tag{16}$$

for some local curvature bound $L(x) > 0$. For brevity, define $r := r(x, \Delta)$. Then

$$\|\delta_f\|_2^2 = \|J\Delta + r\|_2^2 = \|J\Delta\|_2^2 + 2\langle J\Delta, r\rangle + \|r\|_2^2. \tag{17}$$

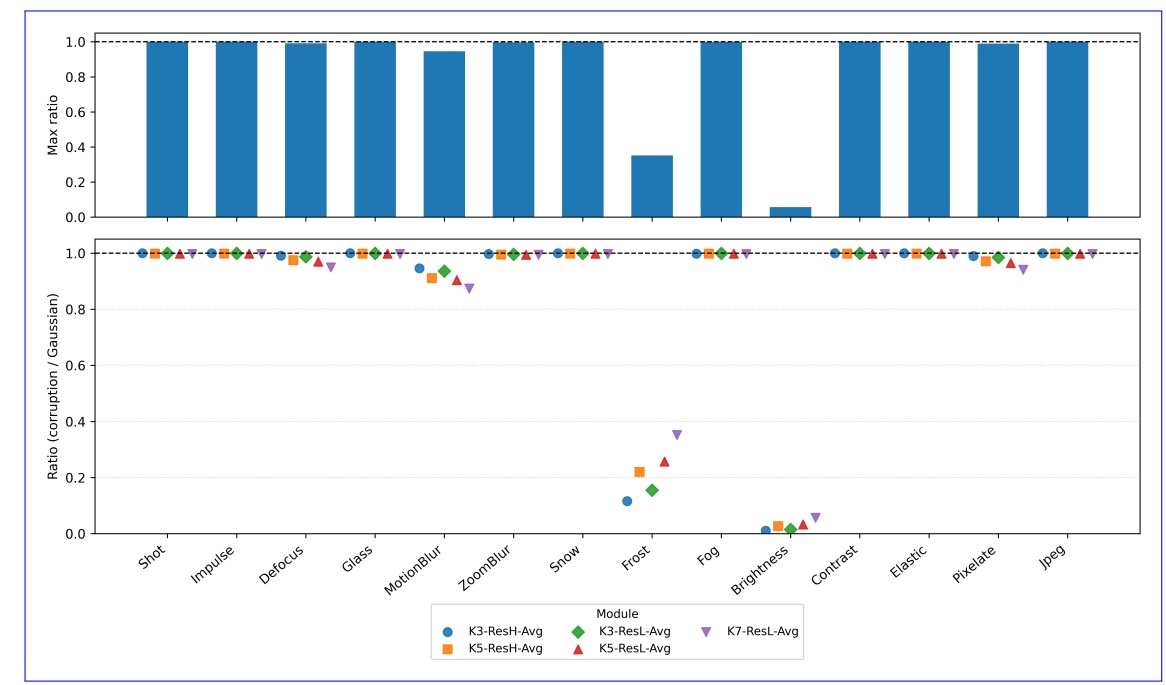

Figure 7: Empirical simulation of Theorem 1, comparing the variance in feature space for common image corruptions and Gaussian noise. All results show the ratio to be close to or less than one, which confirms that Gaussian noise serves as an upper bound.

**Leading term.** Taking expectations and using $\mathbb{E}[\Delta] = 0$ with covariance $\Sigma_\Delta := \mathbb{E}[\Delta\Delta^\top]$, we obtain

$$\mathbb{E}\big[\|J\Delta\|_2^2\big] = \mathbb{E}\big[\Delta^\top J^\top J\Delta\big] = \mathrm{tr}\big(J^\top J\mathbb{E}[\Delta\Delta^\top]\big) = \mathrm{tr}\big(J^\top J\Sigma_\Delta\big) = \mathrm{tr}\big(J\Sigma_\Delta J^\top\big). \tag{18}$$

Now use the spectral constraint $\Sigma_\Delta \preceq \sigma^2 I_d$. Let $B := J^\top J \succeq 0$ and write the eigen-decomposition $\Sigma_\Delta = Q\Lambda Q^\top$, where $\Lambda = \mathrm{diag}(\lambda_1, \ldots, \lambda_d)$ with $0 \le \lambda_i \le \sigma^2$. Then

$$\mathrm{tr}(J\Sigma_\Delta J^\top) = \mathrm{tr}(B\Sigma_\Delta) = \mathrm{tr}(BQ\Lambda Q^\top) = \mathrm{tr}(\Lambda Q^\top BQ)$$

$$= \sum_{i=1}^d \lambda_i (Q^\top BQ)_{ii} \le \big(\max_i \lambda_i\big) \sum_{i=1}^d (Q^\top BQ)_{ii}$$

$$\le \sigma^2 \mathrm{tr}(Q^\top BQ) = \sigma^2 \mathrm{tr}(B) = \sigma^2 \|J\|_F^2. \tag{19}$$

Combining Eq. 18 and Eq. 19 yields

$$\mathrm{tr}\big(J\Sigma_\Delta J^\top\big) \le \sigma^2 \|J\|_F^2. \tag{20}$$

**Remainder terms.** Next, we control the second and third terms in Eq. 17 using the remainder bound in Eq. 16. First, by Cauchy–Schwarz,

$$\big|\langle J\Delta, r\rangle\big| \le \|J\Delta\|_2 \|r\|_2 \le \|J\|_F \|\Delta\|_2 \cdot \tfrac{L(x)}{2} \|\Delta\|_2^2 = \tfrac{L(x)}{2} \|J\|_F \|\Delta\|_2^3, \tag{21}$$

so

$$\big|\mathbb{E}[\langle J\Delta, r\rangle]\big| \le \tfrac{L(x)}{2} \|J\|_F \mathbb{E}\big[\|\Delta\|_2^3\big]. \tag{22}$$

Similarly, from $\|r\|_2^2 \le \frac{L(x)^2}{4}\|\Delta\|_2^4$, we have $\mathbb{E}\big[\|r\|_2^2\big] \le \frac{L(x)^2}{4}\mathbb{E}\big[\|\Delta\|_2^4\big]$.

To interpret the big-$O$ term, it is natural to consider a family of small perturbations $\Delta = \varepsilon\xi$ with a fixed random vector $\xi$ and $\varepsilon > 0$ a scale parameter controlling the perturbation magnitude. Then $\mathbb{E}\|\Delta\|_2^3 = \varepsilon^3\mathbb{E}\|\xi\|_2^3$ and $\mathbb{E}\|\Delta\|_2^4 = \varepsilon^4\mathbb{E}\|\xi\|_2^4$, so we have

$$2\big|\mathbb{E}[\langle J\Delta, r\rangle]\big| + \mathbb{E}\big[\|r\|_2^2\big] = O(\varepsilon^3) = O\big(\mathbb{E}\|\Delta\|_2^3\big), \tag{23}$$

with a constant depending only on $J$, $L(x)$, and the law of $\xi$. We summarize this as $O\big(\mathbb{E}\|\Delta\|_2^3\big)$ in the statement of the theorem.

**Putting everything together.** Taking expectations in Eq. 17 and combining Eq. 18, Eq. 20, and Eq. 23, we obtain

$$\mathbb{E}\big[\|f(x+\Delta) - f(x)\|_2^2\big] = \mathrm{tr}\big(J\Sigma_\Delta J^\top\big) + O\big(\mathbb{E}\|\Delta\|_2^3\big)$$
$$\le \sigma^2\|J\|_F^2 + O\big(\mathbb{E}\|\Delta\|_2^3\big), \tag{24}$$

which proves the first claim.

**Gaussian case and saturation.** Now let $\eta \sim \mathcal{N}(0, \sigma^2 I_d)$. Then $\Sigma_\eta = \sigma^2 I_d$, and the leading term becomes

$$\mathrm{tr}\big(J\Sigma_\eta J^\top\big) = \mathrm{tr}\big(J(\sigma^2 I_d)J^\top\big) = \sigma^2\mathrm{tr}\big(JJ^\top\big) = \sigma^2\|J\|_F^2. \tag{25}$$

The same remainder analysis as above, applied with $\Delta = \eta$, yields

$$\mathbb{E}\big[\|f(x+\eta) - f(x)\|_2^2\big] = \sigma^2\|J\|_F^2 + O\big(\mathbb{E}\|\eta\|_2^3\big). \tag{26}$$

Thus, among all zero-mean perturbations with covariance $\Sigma_\Delta \preceq \sigma^2 I_d$, Gaussian noise $\eta \sim \mathcal{N}(0, \sigma^2 I_d)$ saturates the upper bound on the leading Jacobian-based contribution to the expected feature-space mean-squared error. $\square$

## L  CORRELATION ANALYSIS

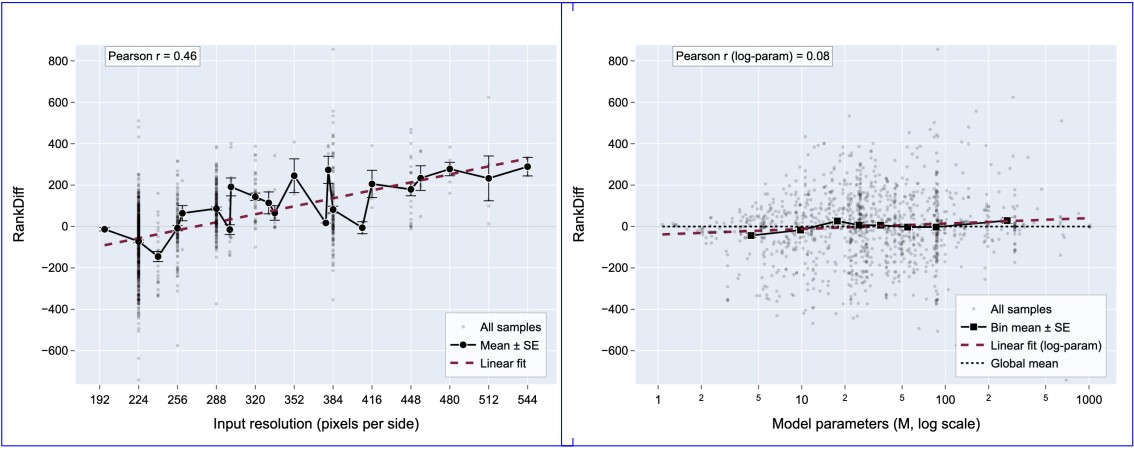

Figure 8: For all 1,174 timm models, we plotted the rank difference with respect to resolution (left) and the number of parameters (right). A significant level of correlation was found for resolution.

The experiments in the main text targeted ViTs and ResNets with a controlled setup for kernel size and resolution. Here, we further found that these observations hold for other vision models. Figure 8 summarizes how the rank difference is affected by resolution and the number of parameters across all 1,174 timm vision models, which also include other models beyond ViTs and ResNets. Firstly, we observed that the number of parameters showed no relationship with the rank difference, which implies that choosing a larger model does not lead to improved robustness against Gaussian noise. By contrast, we observed that the resolution, as well as the kernel size, had a significant level of correlation with the rank difference. Note that this correlation arises even though there are plenty of other factors that affect robustness, such as different training recipes. Overall, smaller resolution led to a smaller rank difference, and this trend holds as a general behavior across vision models.

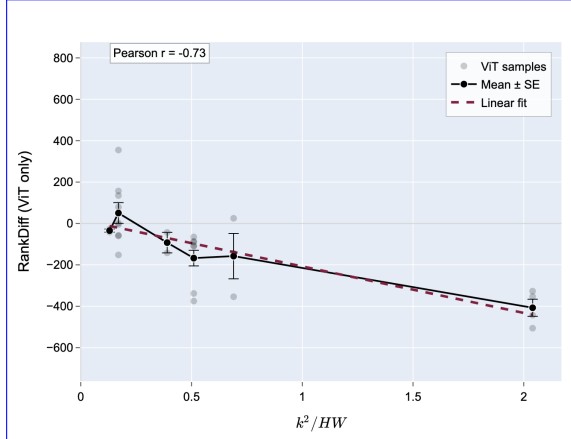

Figure 9: For ViTs, the rank difference exhibited a significant level of correlation with the ratio of patch to image.

Extending the findings of the main text, we can also say that the ratio of patch to image has a significant correlation with robustness against Gaussian noise. For ViTs, this ratio becomes $100 \cdot k^2/HW$ (%). We investigated its relationship with the rank difference (Figure 9), targeting the ViTs listed in Table 1. Again, although these ViTs were trained with different recipes, the overall tendency showed a significant correlation: a higher ratio of patch to image led to a smaller rank difference, indicating improved robustness.

