# OpenReview forum: "Unlocking Noise-Resistant Vision: Key Architectural Secrets for Robust Models"
_ICLR.cc/2026/Conference — Submitted to ICLR 2026_

### Official Review · Reviewer_USA8 · 2025-10-31

**Soundness:** 4
**Presentation:** 4
**Contribution:** 3
**Rating:** 8
**Confidence:** 4

**Summary:**

The paper investigates why certain vision architectures are naturally more robust to additive Gaussian noise and translates these findings into simple, theoretically grounded design principles. The authors evaluate over 1,100 pretrained models and identify four recurring factors that consistently improve robustness: larger stem kernels, smaller input resolutions, average pooling instead of max pooling, and supervised ViTs rather than CLIP ViTs. They complement these large-scale empirical results with theoretical analyses showing that noise attenuation scales quadratically with kernel size and downsampling factor, that average pooling reduces variance while max pooling introduces bias, and that CLIP’s normalization amplifies sensitivity by up to 1.9× compared to standard preprocessing. Altogether, the paper connects architecture-level design choices to quantifiable noise robustness and provides actionable guidelines for building more stable vision models.

**Strengths:**

**Strengths**
- Evaluates over a thousand pretrained models, ensuring strong empirical backing.
- The paper, motivation and insights are clearly written.
- The practical guidelines are directly usable for robust model design and could influence model architecture choices in industry and academia.
- A nice overview that includes mathematical rigor and applied insight.

**Weaknesses:**

**Weaknesses:**
- Could you elaborate more in the differences between the different architecture of ResNet-{C,D,T,S}, maybe highlights the differences with a checkmark system in Table 2 to make the insight of Table 2 clearer because right now the takeaway by looking past the table is not clear.
- The name in the Table 1 of the models is not very easy to parse, if you could make it clearer having column for each component, if would be much easier to compare.
- The study focuses solely on Gaussian noise, which, while analytically convenient, may not capture real-world corruptions (e.g., blur, brightness, compression). Extending to more diverse perturbations would strengthen generality. Or at least add a limitation section mentioning that these findings might affect other perturbations in different ways. Building on that, I then find the title borderline misleading. The findings show some architectural changes to make the model robust to gaussian noise, not robust in general. Same for you conclusion:  "[...] consistently improved * robustness". l. 413 -> Replace: * = "**gaussian noise**" would be less misleading.
So, please tone down some claim.

**Questions:**

**Questions:**

- I am slightly bothered by the fact the the ICLR template format is not the same as the original template, any reason for that ? Line spacing seem different than the other papers.
- Please add a limitation section mentioning the limitations. These architecture changes might work for this specific case, but they are not solution for all corruptions/OOD sources.


Addressing these points (**Questions** and **Weaknesses**) would clarify the analysis, make the results more interpretable, and strengthen the positioning of the paper within the broader robustness literature.

---

> ### Author Response · Authors · 2025-11-17
> **Author Response to Reviewer USA8**
>
> We would like to express our gratitude to the reviewer for the time and efforts to provide valuable comments to improve this manuscript. We find that your comments are worthy of being reflected in our revised manuscript. In this rebuttal and revision, we have done our best to address all your concerns. Here is a point-to-point response.
>
> W1, W2. (Reorganization. See Tables 1 and 2.) Thank you for sharing a great idea. Following your suggestion, we have revised Tables 1 and 2. Specifically, we have restructured Table 1 to explicitly separate model size, patch size, and resolution columns. For Table 2, we also added the difference in ResNet within C, D, T, and S types by specifying detailed configurations such as the configuration in stem conv, width, and downsample. We now feel that the revised tables look far more intuitive to understand the architectural difference.
>
> W3-1. (Clarification. See Section 3 and Appendix K.) Thank you for sharing important points. Indeed, one might consider image corruptions such as ImageNet-C to investigate the robustness of vision models. Regarding these corruptions, we have added Section 3 and Appendix K in the revised manuscript to discuss how our analysis of Gaussian noise can be extended to them. In short, we present several reasons why analyzing robustness to Gaussian noise is a useful proxy that correlates with the robustness under several common corruptions.
>
> W3-2. (Additional Experiments. See Figure 7.) Furthermore, we present additional results on other corruptions in the revised manuscript. Specifically, we performed an empirical simulation to investigate the validity of Theorem 1. Using common image corruptions, including blur, weather, and digital corruptions used in ImageNet C, we first calibrated each corruption and Gaussian noise to have an equal maximum eigenvalue in pixel space and then compared the variance in feature space when passing the same linear stem. Theoretically, Gaussian noise achieves an upper bound on this variance, and our simulations also support this expectation: the ratio of corruption to Gaussian in feature-space variance saturates around 1 across all corruptions (Figure 7). These results clearly demonstrate that the analysis of Gaussian noise captures the worst-case robustness in the image corruptions. We hope that this part will convince you why studying Gaussian noise is useful to these common corruptions and will strengthen the rationale to target the Gaussian noise. Thank you again for pointing out this issue, and please check our revised manuscript.
>
> W3-3. (Clarification. See Title and Conclusion.) We agree with your point that we should tone down the claim of “robustness” to “robustness against Gaussian noise” for clarity. Following your comment, this point has been reflected in the title and conclusion of this manuscript. Note that we found that the title information in OpenReview cannot be updated during the rebuttal and revision period; we will update the title information in OpenReview in the next step.
>
> Q1. (Reorganization.) Thank you for providing a careful checkpoint, and we apologize for any inconvenience. We have corrected the LaTeX template to be consistent with the ICLR 2026 format. Please check the revised manuscript.
>
> Q2. (Clarification. See Appendix F.) Thank you for the suggestion. The limitation section has been presented in Appendix F. We clarify that this part has discussed the points you mentioned, such as the limitation in studying all corruptions and all vision models.
>
> Again, thank you for your kind comments. We now feel that the revised manuscript looks significantly improved, following your comments. We believe that all your concerns can be addressed in this rebuttal and revision. If you have any further concerns, please feel free to share them with us.

---

> > ### Comment · Reviewer_USA8 · 2025-11-22
> >
> > Dear authors,
> >
> > Thank you for your response. I believe that the revisions to the manuscript and the answers provided in the rebuttal adequately address the questions and concerns I previously raised.
> >
> > In its current form, the paper is of good quality. The scope remains somewhat limited to Gaussian noise, but with the updated title and adjusted claims, the manuscript now more accurately reflects this narrower focus. The findings, although restricted to this setting, are still interesting and relevant.
> >
> > I would, however, strongly recommend moving the limitations section from the Appendix into the main paper, since it is an important component that requires greater visibility. Similarly, the newly introduced Appendix M should be referenced in the main text, and its results should be briefly discussed within a dedicated Discussion or Limitations section.
> >
> > Overall, I consider the paper suitable for acceptance. That said, given the relatively narrow scope, I would also understand and not oppose a rejection if other reviewers view this as a significant drawback.

---

> > > ### Author Response · Authors · 2025-12-02
> > > **Author Response to Reviewer USA8 (2)**
> > >
> > > Thank you for hearing from you again. We have uploaded the new revised manuscript, which moved the limitation section into the main text. We have also briefly mentioned Appendix M (now Appendix L) in the main text. Please check out the new revised manuscript. Thank you again for your kind words.

---

### Official Review · Reviewer_RMHn · 2025-11-01

**Soundness:** 2
**Presentation:** 3
**Contribution:** 2
**Rating:** 2
**Confidence:** 3

**Summary:**

In this paper, the authors perform an empirical study on the effect of architectural design on the model robustness against Gaussian noise. The authors evaluate a large number of vision models from timm library and identify four design choices for improved robustness: larger stem kernels, smaller input resolutions, average pooling, and supervised ViTs. The authors also provide the theoretical analysis for each choice.

**Strengths:**

-	The paper is generally well-written and easy to follow.
-	The experiments are comprehensive, covering a large number of models.

**Weaknesses:**

-	The authors mainly focus on Gaussian noise, which makes the scope of the paper somewhat limited. There are many corruptions beyond Gaussian noise. For example, 15 corruptions in [1] are commonly used to evaluate the model robustness. Besides, there are many other robustness benchmarks such as ImageNet-Adversarial, ImageNet-Rendition, ImageNet-Sketch, etc. Broadening the current scope will add more insights to the paper.
-	Since the authors focus on architectures, the mode size is also one important dimension that should be considered, where the current analysis is missing.
-	Apart from ViT and ResNet, there are also many other architectures such as ConvNeXt, Swin, VMamba, Diffusion Classifier, etc. These more recent architectures should be considered and studied as well.

**Reference:**

[1] Benchmarking Neural Network Robustness to Common Corruptions and Perturbations. In ICLR, 2019.

**Questions:**

I am concerned about the questions mentioned above. Given the current status of the paper, I am leaning towards rejection and hope the authors could address my concerns during the rebuttal.

---

> ### Author Response · Authors · 2025-11-17
> **Author Response to Reviewer RMHn**
>
> We would like to express our gratitude to the reviewer for the time and efforts to provide valuable comments to improve this manuscript. We find that your comments are worthy of being reflected in our revised manuscript. In this rebuttal and revision, we have done our best to address all your concerns. Here is a point-to-point response.
>
> W1-1. (Clarification. See Section 3 and Appendix K.) Thank you for sharing important points. Indeed, one might consider image corruptions such as ImageNet-C to investigate the robustness of vision models. Regarding these corruptions, we have added Section 3 and Appendix K in the revised manuscript to discuss how our analysis of Gaussian noise can be extended to them. In short, we present several reasons why analyzing robustness to Gaussian noise is a useful proxy that correlates with the robustness under several common corruptions.
>
> W1-2. (Additional Experiments. See Figure 7.) Furthermore, we present additional results on other corruptions in the revised manuscript. Specifically, we performed an empirical simulation to investigate the validity of Theorem 1. Using common image corruptions, including blur, weather, and digital corruptions used in ImageNet C, we first calibrated each corruption and Gaussian noise to have an equal maximum eigenvalue in pixel space and then compared the variance in feature space when passing the same linear stem. Theoretically, Gaussian noise achieves an upper bound on this variance, and our simulations also support this expectation: the ratio of corruption to Gaussian in feature-space variance saturates around 1 across all corruptions (Figure 7). These results clearly demonstrate that the analysis of Gaussian noise captures the worst-case robustness in the image corruptions. We hope that this part will convince you why studying Gaussian noise is useful to these common corruptions and will strengthen the rationale to target the Gaussian noise. Thank you again for pointing out this issue, and please check our revised manuscript.
>
> W2, W3. (Additional Experiments. See Appendix M.) Thank you for providing valuable comment. In the revised manuscript, we now present additional results. Specifically, we have added results for 1,174 timm models with respect to resolution and the number of parameters (Figure 8). We observed that the resolution exhibited more significance to the robustness against Gaussian noise than that of the model size. This correlation was observed even in the different training recipes. Regarding the model size, however, we did not observe a clear correlation with robustness against Gaussian noise. In fact, one may notice that larger ViTs in Table 1 did not always show improved robustness compared with small or tiny ViTs, whereas larger kernel size and smaller resolution bring consistent improvements. Note that, in Figure 8, we targeted 1,174 timm models, which includes other vision models such as ConvNeXt and Swin, presenting a generic relationship. We hope that the new Figure 8 will convince you of your concerns about the model size factor as well as the extension to other models.
>
> Again, thank you for providing careful comments. We now feel that the revised manuscript looks significantly improved, following your comments. We believe that all your concerns can be addressed in this rebuttal and revision. If you have any further concerns, please feel free to share them with us.

---

### Official Review · Reviewer_dmMx · 2025-11-05

**Soundness:** 2
**Presentation:** 1
**Contribution:** 1
**Rating:** 2
**Confidence:** 3

**Summary:**

This paper tries to study the robustness of different image classification models towards Gaussian noise perturbation. By comparing models' performance rank changes in a big leaderboard before and after adding Gaussian noise to the input images, this paper identifies some architectural designs related to the model robustness. Through analyzing the in-domain performance on different dataset, authors claim that ViTs require larger patch size, smaller input resolution and supervised CE loss to be robust to gaussian noise. For ResNet models, average pooling contributes more to the robustness compared to the max pooling and nearest neighbor pooling. For each conclusion, the paper also presents theoretical analysis as explanations.

**Strengths:**

1. This paper presents a way to analyze the model robustness by looking at the rank difference in the leaderboard before and after applying noise.
2. This paper provides comprehensive theoretical analysis for the observed phenomena.

**Weaknesses:**

1. The organization and the writing for this paper require improvement.
2. There are some significant factual mistakes in the paper, i.e. the authors claim CLIP is a self-supervised method.
3. The conclusion in this paper lacks reasonable support and looks unconvincing. For example, the authors claim larger patch size and lower input resolution image can benefit robustness of ViTs by looking at the performance of existing ViT checkpoints. However, the paper doesn't study the relationship between robustness with patch size and resolution on ViTs. Instead, authors try to verify this conclusion on ResNet backbone. I believe there exists some logic gap between the conclusions and the evidence presented in the paper.
4. This paper only discuss the in-domain setting. It is well known models quite differently for in-domain data and out-of-domain data. Only analyzing model behaviors on in-domain data cannot reflect model's robustness comprehensively.

**Questions:**

Please see the weakness part.

---

> ### Author Response · Authors · 2025-11-17
> **Author Response to Reviewer dmMx**
>
> We would like to express our gratitude to the reviewer for the time and efforts to provide valuable comments to improve this manuscript. We find that your comments are worthy of being reflected in our revised manuscript. In this rebuttal and revision, we have done our best to address all your concerns. Here is a point-to-point response.
>
> W1. (Reorganization.) Thank you for your kind comment. During this rebuttal and revision, we have improved the writing of this manuscript in terms of section organization, table structures, additional clarification, and additional results, which have further enhanced the presentation. If you have any further suggestions to improve the writing, please feel free to share them with us.
>
> W2. (Clarification. See Section 6.1.) We agree with your point. CLIP uses text as an external signal, so it would be correct to mention it as a contrastive learning method rather than a self-supervised learning method. Following your comment, this part has been corrected, and please refer to Section 6.1 of the revised manuscript. We also have carefully checked related descriptions to avoid similar factual inaccuracies.
>
> W3-1. (Additional Experiments. See Appendix M.) Thank you for providing valuable comment. In the revised manuscript, we now present additional results on other vision models as well as ViTs. Firstly, we added results for 1,174 timm models with respect to resolution and the number of parameters (Figure 8). We observed that the resolution exhibited more significance to the robustness against Gaussian noise than that of the model size. This correlation was observed even in the different training recipes. For ViTs, we also added the analysis of how the kernel size and resolution affect the robustness (Figure 9).
>
> W3-2. (Clarification. See Section 4.1.) Additionally, we remind you that Table 1 itself is a controlled experiment on ViTs that allows us to compare the difference between kernel size and resolution and their effects on robustness against Gaussian noise. For example, by comparing vit base patch16 224.augreg in1k and vit base patch32 224.augreg in1k, we can study the effect of the choice of patch sizes of 16 and 32 on performance because all other conditions remained the same. Specifically, we derived the relationship in both ViTs (Table 1) and ResNets (Figure 1). We clarify that this part has been discussed in Section 4.1.
>
> W4-1. (Clarification. See Appendix I.) Thank you for providing a careful checkpoint. Firstly, we clarify that Gaussian noise is applied only during the test phase, not during the training phase. Specifically, in our setup, the training dataset is clean, whereas the test data is noisy, which resembles an out-of-domain setup. We clarify that this part has been discussed in Appendix I.
>
> W4-2. (Clarification. See Section 3 and Appendix K.) Secondly, one might consider out-of-domain data with image corruptions, such as one with ImageNet-C. Regarding these corruptions, we have added Section 3 and Appendix K in the revised manuscript to discuss how our analysis of Gaussian noise can be extended to them. In short, we present several reasons why analyzing robustness to Gaussian noise is a useful proxy that correlates with the robustness under several common corruptions.
>
> W4-3. (Additional Experiments. See Figure 7.) Finally, we present additional results on other corruptions in the revised manuscript. Specifically, we performed an empirical simulation to investigate the validity of Theorem 1. Using common image corruptions, including blur, weather, and digital corruptions used in ImageNet C, we first calibrated each corruption and Gaussian noise to have an equal maximum eigenvalue in pixel space and then compared the variance in feature space when passing the same linear stem. Theoretically, Gaussian noise achieves an upper bound on this variance, and our simulations also support this expectation: the ratio of corruption to Gaussian in feature-space variance saturates around 1 across all corruptions (Figure 7). These results clearly demonstrate that the analysis of Gaussian noise captures the worst-case robustness in the image corruptions. We hope that this part will convince you why studying Gaussian noise is useful to these common corruptions and will strengthen the rationale to target the Gaussian noise. Please check our revised manuscript and tell us whether all these parts would address your concerns on out-of-domain setup.
>
> Again, thank you for your thoughtful comments. We now feel that the revised manuscript looks significantly improved, following your comments. We believe that all your concerns can be addressed in this rebuttal and revision. If you have any further concerns, please feel free to share them with us.

---

### Official Review · Reviewer_iYNm · 2025-11-06

**Soundness:** 3
**Presentation:** 3
**Contribution:** 2
**Rating:** 4
**Confidence:** 4

**Summary:**

This paper studies various vision models to determine architectural choices that are related to higher robustness gaussian noise. The paper finds that larger kernels in the stem network, average pooling and larger normalization standard deviations are related to enhanced robustness.

**Strengths:**

- Extensive experiments
- Mostly clear presentation
- Good theoretical analysis

**Weaknesses:**

- It is not surprising, and known, that larger kernel sizes lead to increased robustness to zero-mean gaussian noise. As the kernel grows larger the (weighted) sum of the noise component in the pixels/features approaches 0.
- Likewise, it is expected that increasing the image resolution, while keeping the kernel size constant, leads to reduced robustness because the number of elements in the downstream feature maps increases, and thus the noise energy in downstream feature map is larger.
    - Perhaps it would be interesting to plot the kernel size as a %age of the image size.
- It is portrayed that unsupervised CLiP is more robust than supervised CLiP, but it is revealed that the only factor is the normalization constant, not the nature of the training objective.
- some clarity issue:
    1. Add gaussian parameters in 3.1
    1. Clarify what the patch size in Section 3 refers to. Is it the longest dimension? Or is it, the total number of pixels in the patch?
    1. In the tables, instead of writing the model name, it would be better to create separate columns for each parameter of the model.
    1. Add details of ResNet-{C,D,T,S} in 4.1.

**Questions:**

If we measure the size of the kernel size as a percentage of the image size, do the observations regarding robustness hold?

---

> ### Author Response · Authors · 2025-11-17
> **Author Response to Reviewer iYNm**
>
> We would like to express our gratitude to the reviewer for the time and efforts to provide valuable comments to improve this manuscript. We find that your comments are worthy of being reflected in our revised manuscript. In this rebuttal and revision, we have done our best to address all your concerns. Here is a point-to-point response.
>
> W1, W2. (Clarification. See Section 4.1.) Thank you for pointing out an important issue. Although the findings on large kernel size and smaller resolution are intuitive and natural, we find that small kernels and large resolutions are still preferred in practice; thus, our claims of choosing a large kernel and small resolution are rather contrary to common practice. For example, the choice in kernel size of ViT is usually 16, not 32, and increasing resolution is preferred in large-scale experiments, such as opting for 384 resolution, not 224. In other words, our study demonstrates that the current practice of adopting small kernels and large resolution rather brings vulnerability against Gaussian noise, which we believe is somewhat surprising to practitioners who have chosen smaller kernels and larger resolution. Furthermore, while the findings on kernel size and resolution might be natural, our contribution is to concretize them via (1) systematically verifying this trend across 1,174 pretrained models and controlled experiments and (2) deriving explicit $k^{-2}$ and $s^{-2}$ scaling laws that quantitatively explain the observed robustness gains. We believe this direction would contribute to both practitioners and the research community. We clarify that this part has been discussed in Section 4.1.
>
> W2-1, Q1. (Additional Experiments. See Appendix M.) Yes, the percentage ratio of patch to image is a valid choice to investigate the robustness of vision models. In the revised manuscript, we now present additional results in Figure 9, where we observed that a higher percentage ratio of patch to image led to smaller rank difference, indicating improved robustness against Gaussian noise. These ratios incorporate the two theories on kernel sizes and resolution and successfully show correlation to robustness against Gaussian noise, even with the different training recipes. Please see Appendix M for a full discussion.
>
> W3. (Clarification. See Section 6.1 and Appendix E.) Thank you for providing a careful checkpoint. In fact, we have performed controlled ablation experiments on CLIP ViTs to investigate possible causes from the differences in training loss, dataset, and architecture. Note that differences in training datasets and losses lead to different pretrained weights for CLIP ViTs. Here, if certain properties of training loss and the dataset are the core cause of vulnerability in CLIP ViTs, we would be able to solve it by swapping certain pretrained weights with those of supervised ViTs. However, it did not solve the vulnerability (Tables 14, 15). Similarly, we also investigated the architectural difference between ViT and CLIP, which was still insufficient to address the vulnerability (Table 16). While there are several possible factors in the difference between CLIP ViTs and supervised ViTs, our finding from these controlled experiments is that the most significant factor among those we analyzed was the choice of normalization constant. We clarify that this part has been discussed in Section 6.1 and Appendix E.
>
> W4-1. (Clarification. See Appendix I.) Thank you for your thoughtful comment. The details on Gaussian parameters have been presented in Appendix I. We clarify that the exact details on Gaussian noise have been precisely explained at the code level.
>
> W4-2. (Clarification. See Section 4.) We apologize for the lack of explanation. It corresponds to the latter: The patch size refers to the size of each patch, following the notation in the original ViT study. In the revised manuscript, we clarified this point by explicitly mentioning that “... a patch size of 16 to set the size of each patch to $16 \times 16$ pixels ...”' in Section 4.
>
> W4-3, W4-4. (Reorganization. See Tables 1 and 2.) Thank you for sharing a great idea. Following your suggestion, we have revised Tables 1 and 2. Specifically, we have restructured Table 1 to explicitly separate model size, patch size, and resolution columns. For Table 2, we also added the difference in ResNet within C, D, T, and S types by specifying detailed configurations such as the configuration in stem conv, width, and downsample. We now feel that the revised tables look far more intuitive to understand the architectural difference. More details on architecture can be found in Appendix I.
>
> Again, thank you for your insightful comments. We now feel that the revised manuscript looks significantly improved, following your comments. We believe that all your concerns can be addressed in this rebuttal and revision. If you have any further concerns, please feel free to share them with us.

---

> > ### Comment · Reviewer_iYNm · 2025-11-25
> >
> > I want to first thank the authors for responding to my comments. I appreciate the time and effort of the authors to make modification to the paper. I still have the following reservations, however, since the author's changed their title to narrow the scope to Gaussian noise, I am slightly increasing my rating.
> >
> > 1. Small kernels and large resolution are preferred because they generally lead to higher accuracy. Perhaps it might be useful to quantify and plot the trade-off in accuracy and robustness as a function of kernel size and resolution. It would be very useful to know if it is possible to achieve robustness gains without sacrificing accuracy by changing kernel size and resolution. This should be easily determined from your extensive experiments. Perhaps you could obtain accuracy scaling laws much like robustness and find an optimal point where accuracy and robustness are balanced.
> > 1. The Gaussian scale is still not mentioned in section 3.1, so its not clear what settings were used to get the results in Table 1. The authors have added this information in the appendix, but, in my opinion, it should be in the main body because it is essential to interpret the results.
> > 1. The rankDiff metric is fine for this paper, but inconvenient for practitioners to use to measure the robustness of their models. Perhaps a better metric might be %-age of flipped predictions. This metric is scale and model invariant, and its computation does not require the accuracy of 1100+ models to be known.
> > 1. I still have doubts about the robustness measurement methodology.
> >     1. The new section 3 justifies the choice of Gaussian noise by arguing that it is the worst case noise in terms of L2-norm/variance. The issue is that it is quiet widely believed that L2-norm is not a good measure of corruptions as far as DNN image classifiers are concerned. For example, translation can induce a very high pixel-wise L2-norm change, however CNNs are translation invariant by design and thus are not expected to suffer greatly as long as the object itself is not occluded. Similarly, the literature on adversarial perturbations has shown that low-norm adversarial attacks can reduce a model's accuracy significantly more than Gaussian noise of much higher variance.
> >     1. It is not clear if the pixel values < 0 and > 1 (or 255) were clipped or retained. If they were retained the images were effectively out of distribution and do not represent a valid input to these models. If they were clipped, then the effective noise scale applied to each image is different.
> >     1. Were the images noised differently for each model, or was each model evaluated on exactly the same noise pattern for each image? If the models were evaluated on different noise patterns, then the results may present some variance under different noise patterns.
> >     1. Even if we take restrict the scope to robustness against L2-norm bounded perturbations, the current methodology of applying a single Gaussian noise perturbation of a random scale to each test image is not rigorous. It would be better if the scale of the noise is fixed.
> >         1. In my opinion, the ideal method would be to estimate the radius of the sphere around test images within which the model's prediction is stable. The larger this radius the more robust a model. See [1] for this method. I understand that this metric is time consuming to compute so its fine if the authors are unable to do it.
> >
> > [1] Cohen, Jeremy, Elan Rosenfeld, and Zico Kolter. "Certified adversarial robustness via randomized smoothing." international conference on machine learning. PMLR, 2019.

---

> > > ### Author Response · Authors · 2025-12-02
> > > **Author Response to Reviewer iYNm (2)**
> > >
> > > Thank you again for providing a careful checkpoint. Firstly, following your comment regarding the noise, we have added more specific details on Gaussian noise and made sure to mention them in the main text of Section 3.1. In short, the pixel values are clipped to resemble a practical scenario, as used in the albumentation library, and we prepared the noisy dataset with fixed noise. Please check out the new revised manuscript.
> > >
> > > We also appreciate the reviewer for continuously sharing new ideas on evaluation. We acknowledge that they are valuable topics and will consider them as future work. Again, thank you for your insightful comments and providing the opportunity to learn a lot from you.

---

### Meta-Review · Area_Chair_gXn5 · 2026-01-13

**Summary:**

Reviewers agree that the paper presents a large-scale and carefully executed empirical study of architectural factors correlated with robustness to Gaussian noise, supported by extensive evaluations and theoretical analysis. However, there is substantial disagreement regarding the significance and scope of the contribution. While one reviewer strongly endorses the work and another views it as a meaningful empirical study after rebuttal, two reviewers raise fundamental concerns about whether robustness to Gaussian noise constitutes a sufficiently broad or meaningful robustness criterion, and whether the resulting conclusions generalize beyond a narrowly defined setting. These concerns persist after rebuttal and indicate a lack of consensus on the paper’s core contribution, which directly informs the decision to reject.

**Reviewer Concerns:**

The rebuttal is careful and addresses many concrete issues, including clarifying experimental setup, correcting factual inaccuracies, adding new experiments, and explicitly narrowing the scope of claims. These changes improve the paper’s clarity and internal consistency. However, the rebuttal does not resolve the deeper disagreement about the relevance and generality of Gaussian noise robustness as a primary evaluation axis. As a result, while the paper is technically sound and responsibly scoped, the rebuttal does not sufficiently alleviate the fundamental concerns raised by multiple reviewers.

**Reviewer Scores:**

Reviewer USA8 gave an initial score of 8 and remained strongly positive after discussion.
Reviewer iYNm gave an initial score of 4 and indicated a willingness to raise their score after rebuttal, plausibly to 6, reflecting improved confidence but continued caution.
Reviewer dmMx gave an initial score of 2 and did not indicate a change in stance after rebuttal.
Reviewer RMHn also gave an initial score of 2 and maintained fundamental concerns about scope and generality, with no signal of score change.

After rebuttal, the submission still has two strong reject scores (2), indicating persistent and fundamental disagreement among reviewers.

---

### Decision · Program_Chairs · 2026-01-26

Reject